



# ALADIN laser frequency stability and its impact on the Aeolus wind error

Oliver Lux[1], Christian Lemmerz[1], Fabian Weiler[1], Thomas Kanitz[2], Denny Wernham[2], Gonçalo Rodrigues[2], Andrew Hyslop[3], Olivier Lecrenier[4], Phil McGoldrick[5], Frédéric Fabre[6], Paolo Bravetti[7], Tommaso Parrinello[8] and Oliver Reitebuch[1]

[1]Deutsches Zentrum für Luft- und Raumfahrt, Institut für Physik der Atmosphäre, 82234 Oberpfaffenhofen, Germany
[2]European Space Agency, European Space Research and Technology Centre, Noordwijk, 2201 AZ, The Netherlands
[3]Vitrociset (a Leonardo company), for ESA, Noordwijk, 2201 DK, The Netherlands
[4]Airbus Defence and Space (Toulouse), Rue des Cosmonautes, 31400 Toulouse, France
[5]Formerly Airbus Defence and Space (Stevenage), Gunnels Wood Rd, Stevenage SG1 2AS, United Kingdom
[6]Les Myriades SAS, Consultancy for Optical Systems, 2 Rue Temponières, 31000 Toulouse
[7]Airbus Italia S.p.A., Via dei Luxardo, 22-24, 00156 Rome, Italy
[8]European Space Agency, European Space Research Institute, 00044 Frascati RM, Italy

*Correspondence to*: Oliver Lux (oliver.lux@dlr.de)

**Abstract.** The acquisition of atmospheric wind profiles on a global scale was realized by the launch of the Aeolus satellite, carrying the unique Atmospheric LAser Doppler INstrument (ALADIN), the first Doppler wind lidar in space. One major component of ALADIN is its high-power, ultraviolet (UV) laser transmitter which is based on an injection-seeded, frequency-tripled Nd:YAG laser and fulfills a set of demanding requirements in terms of pulse energy, pulse length, repetition rate as well as spatial and spectral beam properties. In particular, the frequency stability of the laser emission is an essential parameter which determines the performance of the lidar instrument, as the Doppler frequency shifts to be detected are on the order of $10^8$ smaller than the frequency of the emitted UV light. This article reports the assessment of the ALADIN laser frequency stability and its influence on the quality of the Aeolus wind data. Excellent frequency stability with pulse-to-pulse variations of about 10 MHz (root mean square) is evident for over more than two years of operations in space despite the permanent occurrence of short periods with significantly enhanced frequency noise (>30 MHz). The latter were found to coincide with specific rotation speeds of the satellite's reaction wheels, suggesting that the root cause are micro-vibrations that deteriorate the laser stability on time scales of a few tens of seconds. Analysis of the Aeolus wind error with respect to ECMWF model winds shows that the temporally degraded frequency stability of the ALADIN laser transmitter has only minor influence on the wind data quality on a global scale, which is primarily due to the small percentage of wind measurements for which the frequency fluctuations are considerably enhanced. Hence, although the Mie wind bias is increased by 0.3 m·s⁻¹ at times when the frequency stability is worse than 20 MHz, the small contribution of 4% from all wind results renders this effect insignificant (<0.1 m·s⁻¹) when all winds are considered. The impact on the Rayleigh wind bias is negligible even at high frequency noise. Similar results are demonstrated for the apparent speed of the ground returns that are measured with the Mie and Rayleigh channel of the ALADIN receiver. Here, the application of a frequency stability threshold that sorts out wind observations with variations larger than 20 MHz changes the accuracy of the Mie and Rayleigh ground velocities by less than 0.15 m·s⁻¹.



# 1 Introduction

The launch of ESA's Aeolus mission in August 2018 was an influential event in the history of spaceborne active remote sensing (Stith et al., 2018; Kanitz et al., 2019; Parrinello et al., 2020). Since then, the first Doppler wind lidar in space offers the acquisition of global wind profiles from the ground up to the lower stratosphere which helps to fill observation gaps in the global wind data coverage, particularly over the oceans, poles, tropics, and the Southern Hemisphere (Stoffelen et al., 2020). In this manner, the lack of wind data on a global scale which represented a major deficiency in the Global Observing System

(GOS) (Baker et al., 2014; Andersson, 2018; NAS, 2018) was mitigated, thus contributing to improve the accuracy of numerical weather prediction (NWP) (Straume et al., 2020). On 9 January 2020, the operational assimilation of the Aeolus wind data started at the European Centre for Medium-Range Weather Forecasts (ECMWF), followed by the German, French, and British weather services Deutscher Wetterdienst (DWD), Météo France, and Met Office in May, June, and December 2020, respectively. Recent assessments of the significance of the Aeolus data for NWP have demonstrated statistically positive

impact, especially in the tropics and at the poles, thus providing a useful contribution to the GOS (Rennie and Isaksen, 2020a; Rennie and Isaksen, 2020b; Martin et al., 2020). This was made possible by the identification of and correction for large systematic errors which had strongly degraded the wind data quality in the initial phase of the mission (Kanitz et al., 2020; Reitebuch et al., 2020). Firstly, dark current signal anomalies on single ("hot") pixels of the Aeolus detectors which had led to wind errors of up to 4 m·s$^{-1}$ were recognized and successfully accounted for by the implementation of dedicated calibration

instrument modes (Weiler et al., 2020). Secondly, biases that were found to be closely correlated with small variations in the temperature distribution across the primary telescope mirror (Rennie and Isaksen, 2020a) could be corrected.

Apart from these two major issues, the performance of Aeolus and particularly the precision of the Rayleigh wind results is impaired by the lower than expected atmospheric return signal levels which deviated from end-to-end simulations by a factor of 2.0 to 2.5 already shortly after launch (Reitebuch et al., 2020). The situation was aggravated by a progressive decrease in

emit energy of the laser transmitter during the first year of operation. Consequently, a switch-over to the redundant laser onboard Aeolus was performed in summer of 2019. The second laser showed higher emit energy at a significantly lower decrease rate (Lux et al., 2020a), so that as of February 2021, it provides nearly 70 mJ of pulse energy.

A common characteristic of both lasers is the occurrence of periods with significantly increased frequency fluctuations that was not observed in the same manner during on-ground tests. The laser frequency stability is a crucial parameter for the Aeolus

mission and Doppler wind lidar instruments in general, given the fact that a wind speed of 1 m·s$^{-1}$ introduces Doppler frequency shifts of only a few MHz at the operating ultraviolet (UV) wavelength of the laser. Hence, the frequency fluctuations have to be on the same order to ensure sufficiently low wind errors. Frequency-stable laser emission is also essential for other space lidar technologies like high spectral resolution lidar (HSRL) or differential absorption lidar (DIAL). These techniques will be applied in the upcoming space missions EarthCARE (Earth Clouds, Aerosols and Radiation Explorer) (Illingworth et al., 2015),

ACDL (Aerosol & Carbon Detection Lidar) (Liu et al., 2019) and MERLIN (Methane Remote Sensing Lidar Mission) (Ehret et al., 2017) which are scheduled for launch within the next five years.





In this context, the topic of micro-vibrations and their influence on the stability of spaceborne lasers is highly relevant. Micro-vibrations are defined as vibrations at frequencies beyond ≈1 Hz which cannot be effectively compensated for by the spacecraft attitude and orbit control system (AOCS) (Toyoshima, 2010). They are generated from mechanical moving devices on a satellite and can propagate along the structural panels, thus disturbing the satellite payload and causing spacecraft pointing instability (Dennehy and Alvarez-Salazar, 2018). Current research is primarily focused on the latter effect, especially with regards to optical communication (Chen et al., 2019). However, micro-vibrations can also materialize as translational accelerations which disturb the optical path difference in interferometers and highly sensitive accelerometers in gravimetry missions. The detrimental impact of micro-vibrations on the frequency stability performance of space lidar instruments has not been investigated in detail so far. With regard to Aeolus the main susceptibility to micro-vibrations is related to the alteration of the laser cavity length which leads to frequency fluctuations of the emitted light. For other space lidar missions like IceSat employing the Geoscience Laser Altimeter System (GLAS) (Abshire et al., 2005), Ice-Sat-2 which operates the Advanced Topographic Laser Altimeter System (ATLAS) (Martino et al., 2019) or CALIPSO, deploying the Cloud-Aerosol Lidar with Orthogonal Polarization (CALIOP) (Winker et al., 2006), the requirements in terms of the laser spectral properties are less stringent than for Aeolus and, hence, micro-vibrations play only a minor role for the stability of the laser. Moreover, precise quantification of the frequency stability is a challenging task and only possible for Aeolus thanks to its unprecedented spectrometers which allow to assess this parameter in orbit. For these reasons, literature on laser frequency stability of space lasers and its impact on the data quality of the retrieved products is rather scarce and more related to the preparation of future space missions (Hovis et al., 2008).

This research article aims to provide a comprehensive overview of the frequency stability of the Aeolus laser transmitters and to explore its influence on the quality of the Aeolus wind data. After a short description of the ALADIN instrument and measurement principle (section 2.1), the results from on-ground tests of the laser spectral properties are recapitulated (section 2.2). In section 2.3, the utilization of the Mie receiver channel for the assessment of the laser frequency stability is explained, followed by an introduction of the periods that were analyzed for the present study (section 2.4). Chapter 3 comprises the results of the analysis, starting with a presentation of the ALADIN laser frequency stability over one selected week of the mission (section 3.1). Section 3.2 then elaborates on the correlation of this parameter with the geolocation of the satellite, leading to the detrimental influence of the reaction wheels on the spectral characteristics of the laser (section 3.3) and the identification of micro-vibrations as the most likely root cause (section 3.4). The fourth chapter addresses the question to what extent the Aeolus data quality is diminished by the temporally increased frequency noise. After a short assessment regarding the amount of affected data in different phases of the mission, the impact of the degraded frequency stability on the accuracy and precision of the wind results (section 4.1) as well as on the ground velocities (section 4.2) is evaluated. Finally, a summary and conclusion of the study is provided in chapter 5 together with an outlook to possible improvements in future space lidar missions.



## 2 Methods and datasets

This chapter will provide a brief description of the ALADIN instrument and its operating principle. After summarizing the results from on-ground tests of the laser frequency stability, the approach to assess the frequency stability in-orbit by using the Mie channel is explained. Afterwards, the periods of the Aeolus mission that were selected for analysis are presented in the context of the overall performance the ALADIN laser transmitters over the course of the two years after launch in 2018.

### 2.1 ALADIN configuration and measurement principle

The direct-detection Doppler wind lidar ALADIN onboard Aeolus is composed of a pulsed, frequency-stabilized UV laser transmitter, transmit-receive optics (TRO), a 1.5 m diameter Cassegrain-type telescope and a dual-channel receiver which is sensitive for both molecular and particle backscatter from clouds and aerosols (ESA, 2008; Stoffelen et al., 2005; Reitebuch, 2012, Reitebuch et al. 2018). A schematic diagram of the instrument is illustrated in Figure 1.

The two fully redundant laser transmitters, referred to as flight models A and B (FM-A, FM-B), are switchable by a flip-flop 110 mechanism (FFM). Both lasers are realized as diode-pumped Nd:YAG lasers in master oscillator power amplifier configuration that are frequency-tripled to 354.8 nm emission wavelength (ESA, 2008; Cosentino et al., 2012, Cosentino et al., 2017; Lux et al., 2020a). A nonplanar Nd:YAG ring laser, frequency-locked to an ultra-low-expansion cavity, provides narrowband seed radiation that is fiber-coupled into the folded cavity of the 80-cm long Q-switched master oscillator (MO). The MO cavity length is actively controlled by means of a piezo actuator in order to find the optimal condition for single longitudinal mode 115 operation for each laser pulse (Cosentino et al., 2017). The cavity control scheme is based on the ramp-hold-fire technique (Henderson et al., 1986) which involves the detection of cavity resonances of the injected seed radiation while sweeping the cavity length and firing the Q-switch at the detected resonance position of the piezo actuator. The actual implementation of this technique in the ALADIN MO is capable of achieving a cavity control length of better than a few nanometers. However, it has the drawback of a delay in the millisecond-regime between the detection of the cavity resonance and the laser pulse 120 emission (Trespiti et al., 2017).

The infrared (IR) single longitudinal mode output pulses from the MO (energy: 5 mJ to 10 mJ, FWHM pulse duration: 20 ns, pulse repetition frequency: 50.5 Hz) are amplified in a double-pass pre-amplifier and subsequent single-pass power amplifier which are each realized by side-pumped and conductively cooled Nd:YAG zigzag slabs. The amplification stage boosts the energy of the IR pulses to more than 250 mJ, before they are guided to the harmonic generation stage of the laser. The latter 125 comprises a set of nonlinear lithium triborate (LBO) crystals to generate UV output pulses with a conversion efficiency of about 25%, resulting in an in-flight emit energy in excess of 60 mJ.


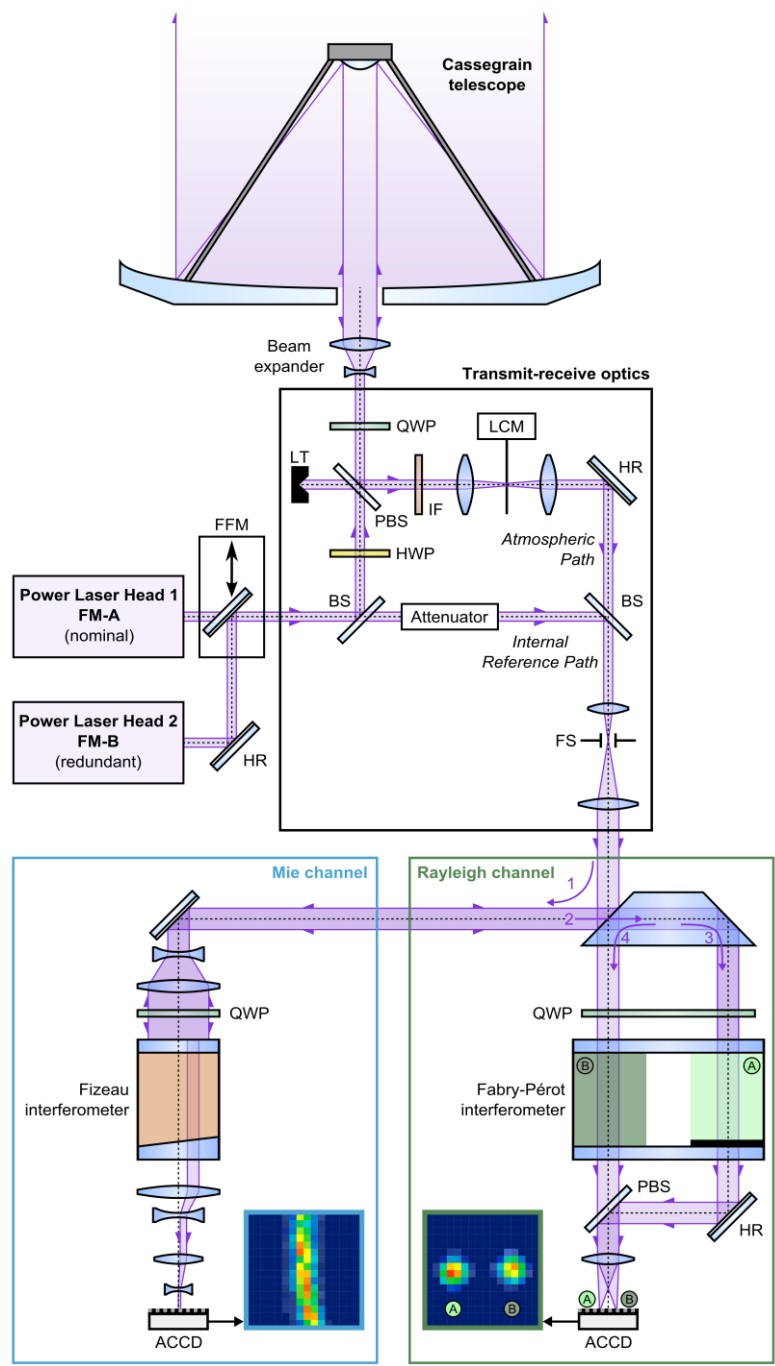

**Figure 1.** Schematic of the direct-detection Doppler wind lidar ALADIN on-board Aeolus. The instrument consists of two fully redundant, switchable UV laser transmitters (FM-A, FM-B), a Cassegrain telescope, transmit-receive optics (TRO) and a dual-channel receiver. The latter is composed of a Fizeau interferometer and sequential Fabry-Pérot interferometers for analyzing the Doppler frequency shift from particulate and molecular backscatter signals, respectively. HR: highly-reflective mirror, FFM: flip-flop mechanism, BS: beam splitter, PBS: polarizing beam splitter, HWP: half-wave plate, QWP: quarter-wave plate, IF: interference filter, LT: light trap, LCM: laser chopper mechanism, FS: field stop, ACCD: accumulation charge coupled device. Numbers indicate the sequential light path in the receiver.





The UV emit beam from one of the two switchable laser transmitters is then directed to the telescope which is used in monostatic configuration, i.e. signal emission and reception are realized via the same primary and secondary mirror. A small portion (0.5 %) of the radiation is separated at a beam splitter (BS) within the TRO configuration and, after being attenuated, guided to the instrument field stop (FS) and receiver channels. This portion is referred to as internal reference path (INT) signal and serves the determination of the frequency of the outgoing laser radiation as well as the calibration of the frequency-dependent transmission of the receiver spectrometers. The INT signal is thus essential for the wind measurement principle of ALADIN which relies on detecting frequency differences between the emitted laser pulses and those backscattered from the atmospheric particles and molecules moving with the ambient wind. The frequency shift $\Delta f_{\text{Doppler}}$ in the backscattered signal that is introduced by virtue of the Doppler effect is proportional to the wind speed $v_{\text{LOS}}$ along the laser beam LOS according to $\Delta f_{\text{Doppler}} = 2 f_0 v_{\text{LOS}}/c$, where $c$ is the speed of light and $f_0$ is the frequency of the emitted light. The atmospheric return signal collected by the telescope enters the TRO where it passes through a laser chopper mechanism (LCM) before being spatially overlapped with the INT beam by another beam splitter. Due to the long travel time of the atmospheric return of a few ms, the detection is temporally separated from the INT signal.

The ALADIN receiver consists of two complementary channels which individually derive the Doppler frequency shift from the narrowband (FWHM ≈ 50 MHz) Mie backscatter from particles like clouds and aerosols on the one hand and from the broadband (FWHM ≈ 3.8 GHz at 355 nm and 293 K) Rayleigh-Brillouin backscatter from molecules on the other hand. The Mie channel is realized by a Fizeau interferometer and relies on the fringe-imaging technique (McKay, 2002) where a linear interference pattern (fringe) is vertically imaged onto the detector. The spatial location of the fringe changes approximately linearly with frequency of the incident light so that a Doppler frequency shift manifests as a spatial displacement of the fringe centroid position. Derivation of the Doppler shift from the broadband Rayleigh-Brillouin backscatter spectrum is achieved by two sequential Fabry-Pérot interferometers (FPIs) that are utilized for applying the double-edge technique (Chanin et al., 1989; Garnier and Chanin, 1992; Flesia and Korb, 1999). The two FPIs act as bandpass filters with adequate width and spacing that are symmetrically placed around the frequency of the emitted laser pulse. Measurement of the contrast between the signals transmitted through the two filters allows to accurately determine the frequency shift between the emitted and backscattered laser pulse.

The Mie and Rayleigh signals are finally detected by two accumulation charge-coupled devices (ACCDs) with an array size of 16 pixels×16 pixels in the imaging zone of the CCD. The electronic charges of all 16 rows in the image zone are then binned together to one row. For the wind profile, this row is transferred to a memory zone. The latter can store 25 rows each representing one vertical range gate of the measured wind profile. It is important to note that the atmospheric signals from 18 successive laser pulses are accumulated already on-board the spacecraft to so-called "measurements" (duration 0.4 s). Since February 2019, $P = 19$ laser pulses are emitted per measurement interval from which one is lost due to the read-out of the ACCD ($P - 1$ setting). Post processing on-ground can sum the signals from 30 measurements, i.e. 540 laser pulses, and form one "observation" (duration 12 s). In contrast to the atmospheric return, the INT signal is acquired for each single pulse and stored directly as one row in the memory zone.





According to the above equation for the Doppler frequency shift, a LOS wind speed of 1 m·s$^{-1}$ translates to a frequency shift of 5.63 MHz (at $f_0$ = 844.75 THz, all subsequent frequency stability values given at this UV frequency). Therefore, in order to

measure wind speeds with an accuracy of 1 m·s$^{-1}$, the required relative accuracy of the frequency measurement is on the order of 10$^{-8}$ which poses stringent requirements on the frequency stability of the laser transmitter. This is especially true, as the atmospheric backscatter signals from multiple outgoing laser pulses are accumulated to measurements before data down-link. This implies that a pulse-to-pulse normalization of the return signal frequency with the internal reference frequency is not possible. Instead, the determined frequency of the pulses averaged over one measurement is affected by inhomogeneous cirrus

or broken clouds, as only a subset of the 18 pulses may be detected in the atmospheric path. As a result, large frequency variations on measurement level (over periods of 0.4 s) in combination with atmospheric inhomogeneities results in significant wind errors, that are then vertically correlated below the (cloud) bin that filters out some of the emitted pulses (Marksteiner et al., 2015). The same holds true for ground return signals from pulses within one measurement that are reflected off a surface with varying albedo, so that the different weighting of the return signals in the accumulation of the pulses results in an error

of the determined ground velocity.

## 2.2 On-ground assessment of the laser frequency stability

Measurement of the ALADIN laser absolute frequency and its temporal stability was done during on-ground tests under vacuum conditions by means of an external wavelength meter (High Finesse WSU10) that was calibrated using a helium neon laser (Mondin and Bravetti, 2017). The experimental setup was provided by DLR and represents an integral part of the

diagnostics for determining the spectral properties of the ALADIN Airborne Demonstrator (A2D) (Lemmerz et al., 2017). Using it for characterization of the ALADIN laser, it was found that the laser frequency stability was well within the specification requirement which states that the root mean square (RMS) variation of the frequency stability over 14 s should be below 7 MHz. The 14 s time period was chosen to ensure adequate SNR over one Aeolus observation (12 s). In addition, sensitivity tests with a thermal cycle of ±0.2°C were carried out for about four hours: one at ambient temperature, one with the

laser interface at +35°C and two at -2.5°C. For all four thermal cycles, the absolute laser frequency varied by less than 25 MHz (peak to peak), demonstrating a good stability also over medium time scales, as thermal variations over one week in orbit were expected to be even below 0.2°C. However, the test also revealed that operation in a vibrational environment, in this case introduced by vacuum pumps and a chiller, led to a degraded laser frequency stability with pulse-to-pulse fluctuations above 30 MHz RMS (Mondin and Bravetti, 2017). Further on-ground test results are discussed in the context of micro-vibrations in

section 3.4.

## 2.3 Utilization of the Mie channel as wavelength meter in orbit

An external wavelength meter is not available in space. However, the spectrometer data gained from the Fizeau interferometer of the Mie channel can be exploited for deriving the spectral properties of the narrowband laser emission. For this purpose, the INT signal that is usually analyzed after accumulation of multiple pulses to measurement level and contained in the L1A

Aeolus data product (Reitebuch et al., 2018) is evaluated for each individual pulse. As stated above, the Mie channel response
is represented by the centroid position of the fringe that is imaged onto the Mie detector and then integrated over the 16 lines
of the ACCD. Figure 2 shows the vertically summed ACCD counts over the 16 horizontal pixels for one laser pulse. The fringe
centroid position is calculated by a Nelder-Mead Downhill Simplex Algorithm (Nelder and Mead, 1965) to optimize a
Lorentzian line shape fit of the signal distribution (Reitebuch et al., 2018). In this manner, the Mie response is derived with

high accuracy. Conversion of the Mie response into relative laser frequency is based on dedicated in-flight calibrations of the
Mie channel from which a sensitivity of ≈100 MHz/pixel was determined. The accuracy of the "inherent wavemeter" on-board
Aeolus is estimated to be 1 MHz. It is mainly limited by the shot-noise-limited signal-to-noise ratio (SNR) of the Mie signal
and the full width at half maximum (FWHM) of the Fizeau interference fringe (≈1.3 pixels). This could be demonstrated with
A2D laboratory measurements comparing the spectrometer response from the ALADIN pre-development model Fizeau

interferometer and parallel wavemeter measurements using the A2D UV laser (Lux et al., 2019). Hence, frequency fluctuations
of the internal path signal can be measured as variations in the calculated Mie response on a pulse-to-pulse basis which allows
to assess the frequency stability of the Aeolus laser transmitter over short- and long-term time scales.

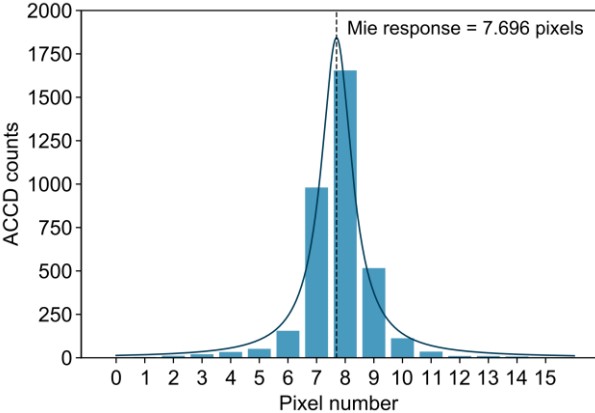

**Figure 2.** Internal reference path Mie signal for one laser pulse (24/11/2020, 0:23:47 UTC). After vertical integration of imaged fringe on
the Mie ACCD (see also Fig. 1), the signal is distributed over 16 pixels (blue bars). A Nelder-Mead Downhill Simplex Algorithm is applied
to determine the centroid position from a Lorentzian line shape fit (dark blue line).

### 2.4 Analyzed periods of in-orbit Aeolus datasets

The laser frequency stability was analyzed for different periods of the Aeolus mission. Since the satellite is circling around the

Earth with an orbit repeat cycle of one week, it was decided to study the performance over selected seven-day periods. This
approach was also motivated by the observed correlation between the frequency stability and the satellite's geolocation (see
section 3.2). After one week the maximum coverage of the globe is reached and the operation timeline, particularly the attitude
control sequence of the platform approximately repeats. In total, five weeks between December 2018, when the mission was
still in commissioning phase after its launch on 22 August 2018, and October 2020 were chosen for investigation, as listed in

Table 1. The table contains information on the operated laser transmitter as well as on the MO output energy (IR), the laser



emit energy (UV) and the laser frequency stability which will be discussed later in the text. The energy values represent the respective mean values and standard deviations from laser internal photodiode readings. It should be noted that the IR energy reported by the MO photodiode of the FM-A is considered inaccurate. Here, Q-switch discharges influence the energy monitoring, as they result in IR radiation with the wrong polarization that is circulating in the MO. This light is partially

incident on the PD, thus corrupting the energy measurement. As can be seen from the table, the UV emit energy of FM-A decreased significantly between December 2018 and May 2019. The degradation of the laser performance was traced back to a progressive misalignment of the MO (Lux et al., 2020a) and led to the decision to switch to the second laser FM-B. This was necessary to ensure a sufficient SNR of the backscatter return and thus a low random error of the wind observations. The FM-B not only delivered a higher initial energy after switch-on in late June 2019 (67 mJ compared to 65 mJ after FM-A switch-on),

but also has been showing a much slower power degradation. After an initial drop by 6 mJ between July 2019 and October 2019, the UV emit energy has decreased by less than 0.08 mJ per week. Thanks to an optimization of the laser cold-plate temperatures in March 2020 which increased the UV energy by about 4 mJ, the energy has remained above 60 mJ since then. Despite the better overall performance compared to the FM-A, the FM-B behavior was observed to be more affected by orbital and seasonal temperature variations of the satellite platform that were transferred to the laser optical bench. As a consequence,

laser anomalies associated with larger energy variations of about ±1 mJ occurred both on shorter (hours) and longer-term time scales (weeks to months).

The periods listed in Table 1 are chosen such that they represent different phases of the mission: the early FM-A phase in December 2018 when the instrument parameters had settled after launch, but the degradation of the MO was already ongoing; the late FM-A phase in May 2019 when the degradation had progressed; the early FM-B phase in October after completed

thermalization of the second laser and later FM-B periods in August and September/October 2020 after optimization of the laser cold-plate temperatures. The latter two periods were also chosen to identify the variability of the frequency stability performance decoupled from the power performance of the laser which was stable in summer and autumn 2020 and free of temperature-related laser anomalies as stated above.

**Table 1.** Overview of the periods that were studied in terms of the laser frequency stability. The active laser transmitter operated in the respective period (FM-A or FM-B) and the corresponding master oscillator IR output energy and UV emit energy (as reported by the laser internal photodiodes) are provided. Note that the reading of the MO photodiode for FM-A is considered inaccurate (see text). The frequency stability is given as the mean of the standard deviations from all wind observations of the respective week.

| Period | Laser transmitter | MO energy | UV emit energy | Frequency stability |
|---|---|---|---|---|
| 17/12/2018 - 24/12/2018 | FM-A | $(8.3 \pm 0.5)$ mJ | $(56.2 \pm 0.8)$ mJ | 7.2 MHz |
| 13/05/2019 - 20/05/2019 | FM-A | $(8.9 \pm 0.3)$ mJ | $(44.2 \pm 0.8)$ mJ | 10.7 MHz |
| 14/10/2019 - 21/10/2019 | FM-B | $(5.87 \pm 0.03)$ mJ | $(60.8 \pm 0.8)$ mJ | 8.1 MHz |
| 17/08/2020 - 24/08/2020 | FM-B | $(5.91 \pm 0.03)$ mJ | $(61.8 \pm 0.3)$ mJ | 8.7 MHz |
| 28/09/2020 - 05/10/2020 | FM-B | $(5.88 \pm 0.03)$ mJ | $(61.4 \pm 0.4)$ mJ | 8.6 MHz |



## 3 Results

The frequency stability of the ALADIN laser will first be presented at one example, namely the week in October 2019, to illustrate the main temporal characteristics of the spectral behavior as well as the relation to the geolocation of the satellite. This leads to the correlation of the laser frequency stability with platform parameters, particularly the reaction wheel speeds which is elaborated on subsequently. This correlation is additionally analyzed for the FM-A period in May 2019 to allow for a comparison between the two flight model lasers. The chapter concludes with a discussion of micro-vibrations as the root

cause of the frequency noise.

### 3.1 Frequency stability of the ALADIN laser transmitters

Figure 3(a) depicts a typical time series of the laser frequency on pulse-to-pulse level over about one orbit that was measured for FM-B on 14 October 2019 between 1:27 UTC and 2:57 UTC. The plot contains the calculated Mie responses from 243,000 pulses distributed over 450 observations of 12 s each. The mean Mie response is 7.25 pixels, corresponding to a fringe centroid

position close to the center of the ACCD, as shown in Fig. 2. The response variations are converted into relative frequency fluctuations considering the sensitivity of the Mie channel of ≈100 MHz/pixel, whereby the mean response value was identified as reference frequency. An important parameter is the standard deviation of the relative frequency over one observation (540 pulses, 12 s), as it describes the frequency stability within those periods that are considered for Rayleigh wind retrieval on observation scale. The mean of the standard deviations from all observations of the 90-minute period accounts for 8.4 MHz

which is comparable to the performance during on-ground tests (section 2.2) and only slightly above requirements (7 MHz). However, it is worse than the in-flight performance of the A2D (<4 MHz RMS) during airborne operations which is characterized by a very similar laser design (Lemmerz et al., 2017, Lux et al., 2020b), but using a more agile ramp-fire technique to actively control the MO cavity length. The lower frequency stability can be ascribed to the fact that the time series also exhibits numerous sporadic periods at which much higher frequency fluctuations occur. During these periods, which last

from a few seconds to several minutes, the variations increase to more than 20 MHz with peak-to-peak variations of up to 150 MHz (see Fig. 3(b)). Despite the periods of increased frequency jitter, the stability of less than 10 MHz is unrivalled by any space-borne high-power laser so far. It should also be noted that frequency variations of 3 MHz (corresponding to 1 MHz of the IR output of the MO), requires a cavity length stabilization of only 3 nm within one oscillator free spectral range of 532 nm or 180 MHz, considering the 80 cm resonator length.

It should be pointed out that, apart from laser frequency variations caused by cavity length changes, the measured Mie response can, in principle, also be altered by angular variations of the laser beam incident on the Fizeau interferometer. In order to estimate the contribution of angular variations, a potential correlation of the Mie response with the far-field beam position was investigated. For this purpose, the horizontal position of the two spots from the internal reference signal that are imaged onto the Rayleigh ACCD (see also Fig. 1) were analyzed during periods of enhanced noise. The studies showed a small spot motion

correlated with the Mie response which is most likely due to the influence of the Fizeau reflection that is promoted by the





sequential configuration of the two receiver channels. The change in spot position was in line with the motion that was observed at times when the laser frequency was deliberately tuned, e.g. during instrument calibrations. This result strongly suggested that the contribution of angular variations to the Mie response fluctuations is negligible and that variations of the internal path Mie response are largely due to changes in the laser frequency.


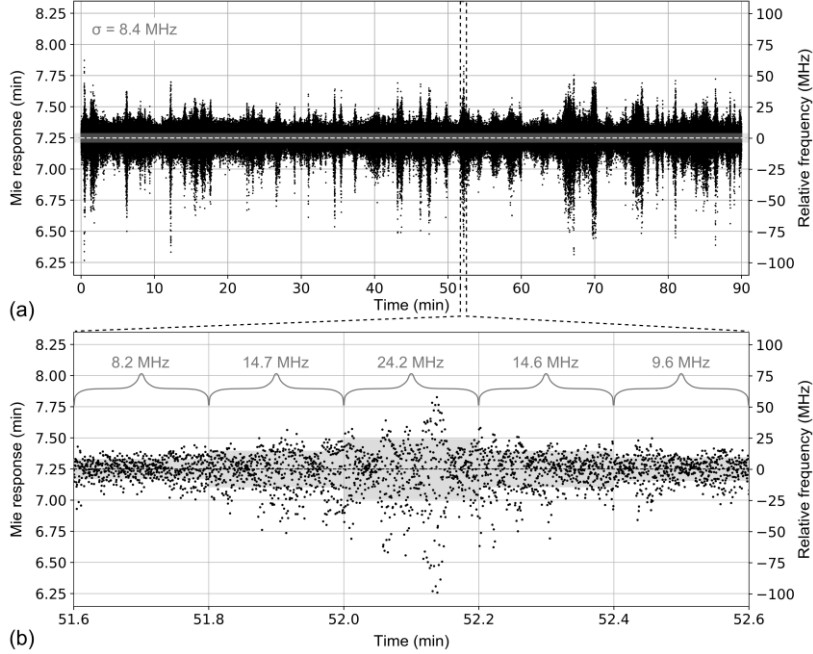

(a)

(b)

**Figure 3.** ALADIN laser frequency stability: (a) Time series of the laser frequency fluctuations over about one orbit (14/10/2019, 1:27 UTC to 2:57 UTC), as measured from the Mie channel of the receiver; (b) Zoom into a selected period showing the frequency variations over five observations. The given values denote the standard deviations σ of the relative frequency on pulse-to-pulse level over the individual observations (540 pulses). The standard deviations are also indicated as grey-shaded areas.


Analysis of the entire week from 14 October 2019 to 21 October 2019 reveals a mean frequency stability of 8.1 MHz over the 49209 observations, as depicted in Fig. 4(a) where the standard deviation is plotted for each observation of the regarded period. Data gaps in the timeline are due to special operations that are regularly performed in each week, such as Instrument Spectral Registration (Reitebuch et al., 2018), Down Under Dark Experiment (Weiler et al., 2020) or orbit correction maneuvers. The

figure also illustrates the percentage of observations that are affected by enhanced frequency fluctuations. While the frequency stability is better than 15 MHz for the vast majority of observations (about 93%), it is worse than 20 MHz for 2.4% and even worse than 25 MHz for almost 1% of the observations. Nevertheless, there is also a considerable amount of observations (19%) for which the frequency stability better than 5 MHz, i.e. comparable to the A2D laser performance.

To estimate the potential impact of the enhanced fluctuations on the wind accuracy, the following calculation is performed.

According to the above equation for the Doppler shift, a frequency difference of 10 MHz is introduced by a line-of-sight (LOS) wind speed of about 1.8 m·s$^{-1}$. Taking into account the off-nadir angle of Aeolus of 37° at the location of the measurement


track, this translates to a horizontal LOS (HLOS) wind speed of 1.8 m·s⁻¹ / sin(37°) ≈ 3.0 m·s⁻¹. Consequently, wind errors of several m·s⁻¹ are potentially caused when only a small subset of emitted pulses from one observation (with σ = 10 MHz) is backscattered from the atmosphere or the ground and contributes to the return signal in the wind retrieval.

Interestingly, the distribution of measured Mie responses (see Fig. 4(b)) indicates that the frequency fluctuations are not symmetrically distributed. Instead, the frequency tends to jump to lower values (= lower responses). This behavior is also visible in Fig. 3, where the largest departures from the mean are negative. As a result, the higher standard deviation over one particular observation is, the larger is the negative shift of the respective mean from the mean over all observations. This relationship is shown in Fig. 4(c) and can most likely be traced back to disturbances of the active stabilization of the MO cavity

length during the periods of enhanced frequency jitter which results in frequency jumps preferentially in one direction. For instance, such jumps occur when the interference signal produced by the seed laser circulating in the MO features parasitic peaks that are erroneously detected as MO cavity resonances. A similar behavior was observed for the A2D in highly vibrational environment or in case of imperfect alignment of the MO.

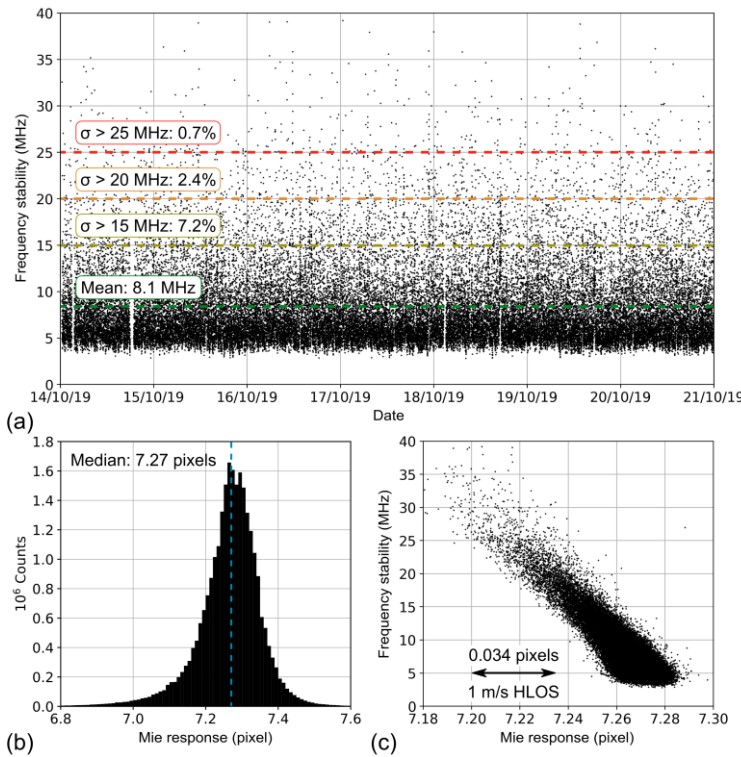

**Figure 4.** (a) Time series of the laser frequency stability over one week from 14 October 2019 to 21 October 2019, i.e. during FM-B operation. The data points represent the standard deviation of the relative frequency on pulse-to-pulse level over one observation (540 pulses). Data gaps are due to calibrations or orbit maintenance. The horizontal lines indicate mean over all observations of 8.1 MHz (green) as well as thresholds to indicate the percentage of observations for which the frequency stability is worse than 15 MHz (yellow), 20 MHz (orange) and 25 MHz (red). (b) Probability density distribution of the Mie internal reference response for the dataset in October 2019. (c) Relationship

between the laser frequency stability in terms of the standard deviation on pulse-to-pulse level over one observation and the mean Mie internal reference response per observation. A response shift by 0.034 pixels translates to a frequency shift by about 3.4 MHz, corresponding to a HLOS wind speed change of 1 m·s⁻¹, assuming a constant response of the atmospheric or ground return signal.





## 3.2 Correlation with the satellite's geolocation

The enhanced frequency fluctuations of the laser transmitter were detected very early in the mission and attributed to potential
vibrations introduced by the satellite platform which affects the MO cavity length (Lux et al., 2020a). However, a correlation
to platform parameters, particularly the rotation velocity of the satellite's reaction wheels was not found initially. This was
mainly due to the fact that only short timelines were analyzed, typically covering only several minutes to hours, as shown in
Fig. 3. Since the platform parameters vary slowly over the orbit, a relationship to the fast changes in the laser frequency
stability within several seconds was considered unlikely.

During the first year of operation, the assessment of the frequency stability was then focused on the weekly Instrument
Response Calibrations (IRCs, Reitebuch et al., 2018). IRCs are required to determine the relationship between the Doppler
frequency shift of the backscattered light, i.e. the wind speed, and the response of the Rayleigh and Mie spectrometers. The
procedure involves a frequency scan over 1 GHz in steps of 25 MHz to simulate well-defined Doppler shifts of the atmospheric
backscatter signal within the limits of the laser frequency stability. During the IRC which takes about 16 minutes, the
contribution of (real) wind related to molecular or particular motion along the instruments' LOS is virtually eliminated by
rotating the satellite by an angle of 35°. This results in nadir pointing of the instrument and, in case of negligible vertical wind,
vanishing LOS wind speed. The IRCs are preferably carried out over regions with high surface albedo in the UV spectral
region, e.g. over ice, to ensure strong return signals, and in turn, high SNR. This is particularly important for the Mie response
calibration which relies on measuring the spectrometer response from the ground return to be then used for the retrieval of
atmospheric winds.

The laser frequency stability was studied for each of the 60 IRCs conducted between 7 September 2018 and 9 December 2019;
most of them over Antarctica, while IRCs #31 to #43 were performed over the Arctic. Here, it was found that the frequency
stability was degraded during the IRCs compared to operation in nominal wind velocity mode (WVM). While it was on the
order of 8 to 10 MHz in WVM, the mean stability over the 16-minute IRC period accounted for 12 to 14 MHz when the
satellite was pointing nadir, suggesting an influence of the platform attitude on the laser. This conclusion was strengthened by
the circumstance that the laser temperatures and energies were strongly varying during the nadir and off-nadir slews before
and after the IRC, respectively. Moreover, these orbit maneuvers involved thruster firings which also caused increased
frequency noise during the slews before and after the weekly IRC, thus pointing to mechanical disturbances as the root cause.
Furthermore, the frequency stability showed to be significantly better over Antarctica ((11.6 ± 1.7) MHz) than over the Arctic
((15.2 ± 2.1) MHz), although the IRC procedure was the same over both locations. Finally, it was noticed that the progression
of the relative laser frequency featured recurring temporal patterns for those IRCs that were carried out over the same locations
in subsequent weeks. For instance, an accumulation of periods with increased frequency jitter were observed at the beginning
of the time series for the Antarctica IRCs, whereas numerous high-noise periods, distributed over the entire procedure, were
evident for the Arctic IRCs. Figure 5 depicts the laser frequency variations over selected IRC periods over the two different
geolocations, clearly demonstrating the reproducibility of the jitter patterns for the weekly IRCs.



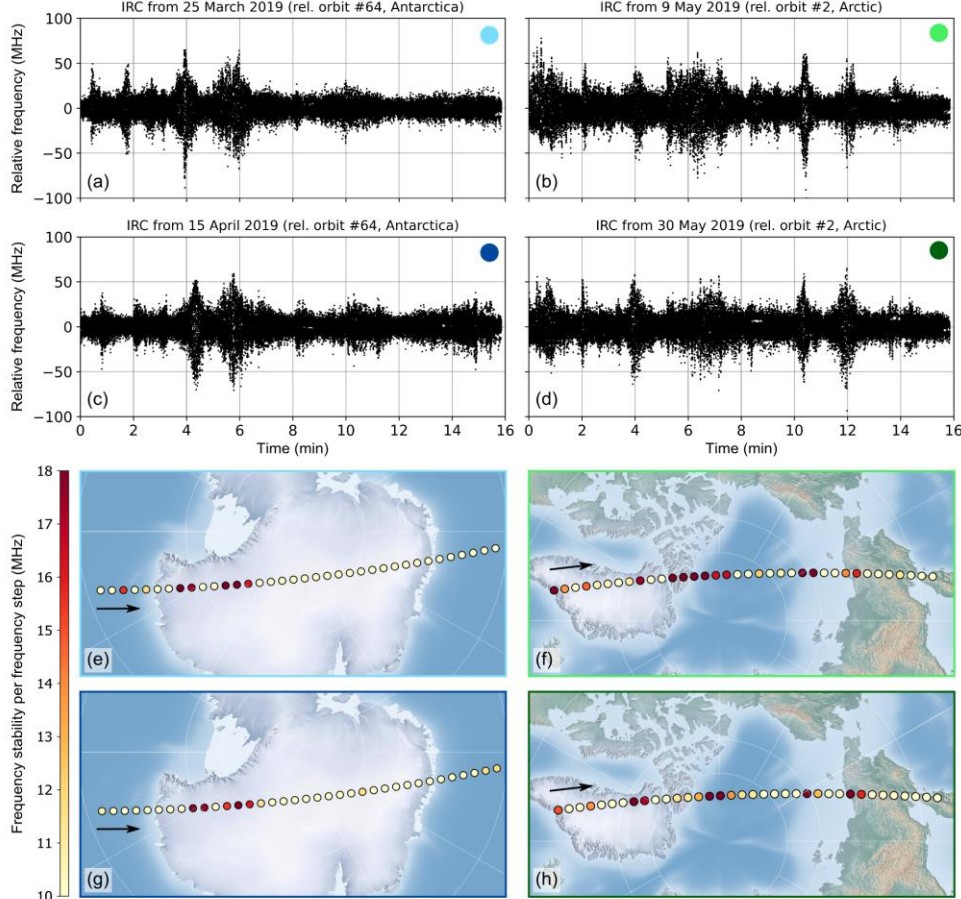

**Figure 5.** Time series of the laser frequency fluctuations over periods of selected IRCs over Antarctica (a, c) and the Arctic (b, d) together with the corresponding geolocations of the individual frequency steps in panels (e) – (h). Each dot corresponds to one frequency step (24 s), whereby the color-coding describes the standard deviation of the relative frequency on pulse-to-pulse level within this period.

Following these observations, the laser frequency stability was studied over one-week periods to review the influence of the satellite's geolocation (see section 2.4). The performance from the week between 14 October and 21 October 2019 (Fig. 4a), based on the Mie response data from more than 27 million laser pulses, is illustrated in Fig. 6. Each dot in the two maps represents one observation, whereby the color and opacity indicate the frequency stability in terms of the standard deviation of the relative frequency over the 540 pulses within that observation. The analysis revealed that the enhanced frequency noise

does not occur randomly, but is correlated with the location of the satellite over the Earth's surface. More strikingly, different linear and circular structures are evident for ascending and descending orbits, which makes clear that the frequency stability does not only depend on the geolocation, but also on the satellite's orientation along the orbit. For ascending orbits those observations with enhanced frequency fluctuations are accumulated in several bands wrapping around the globe, most prominently around East Asia and the Pacific Ocean. In contrast, multiple narrow latitudinal bands and a circular structure

over the South Atlantic and South Indian Ocean are apparent for descending orbits.

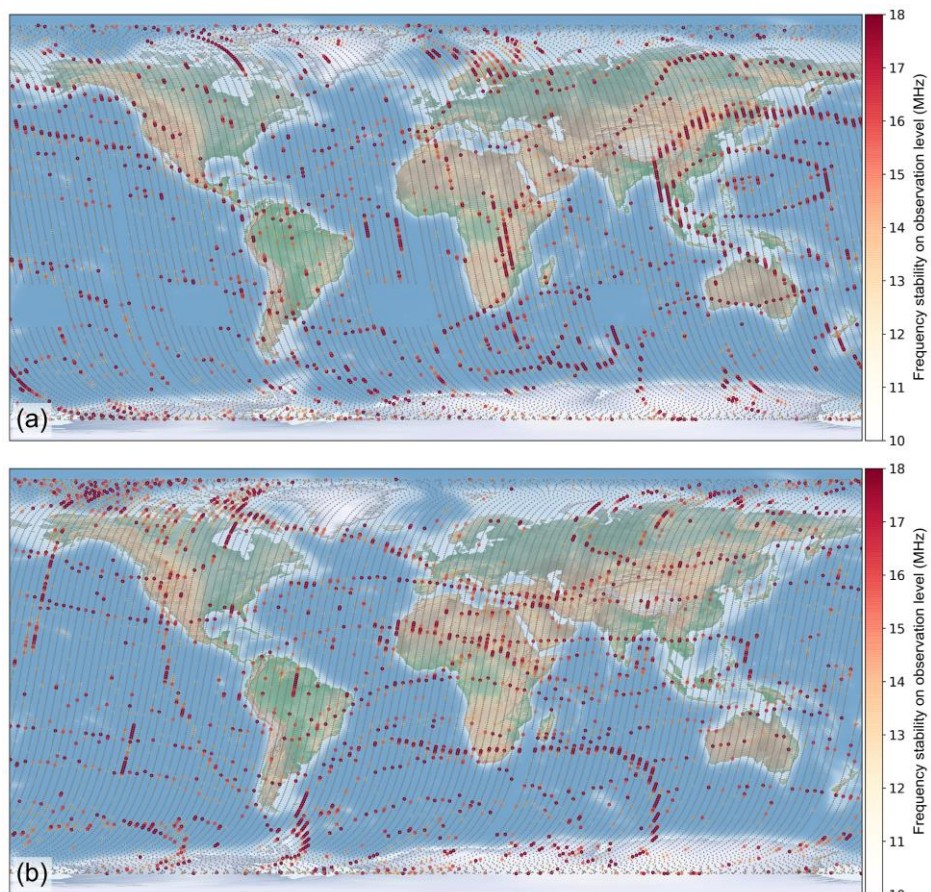

**Figure 6.** Geolocation of wind observations with enhanced frequency noise for (a) ascending and (b) descending orbits. The plot contains the data from the week between 14/10/2019 (0 UTC) and 21/10/2019 (0 UTC) (see also Fig. 4). Each dot corresponds to one observation (12 s), whereby the color-coding describes the standard deviation of the relative frequency on pulse-to-pulse level within this period. Note that the opacity of the dots also scales with the frequency stability so that observations with σ < 10 MHz are not visible.

The geolocational patterns were found to be reproducible for the investigated FM-B periods with only slight variations (<200 km). This is especially true for the linear and circular structures that are formed by neighboring orbits, i.e. with large temporal distance. On the contrary, continuous phases of enhanced frequency noise over several observations which manifest as orange and red lines along one orbit, e.g. in Northern Canada or over Northern Europe in Fig. 6(a), were evident at different geolocations for the weeks in August 2020 and September/October 2020.

Similar correlation of the laser frequency stability with the satellite position was also obvious for the periods in December 2018 and May 2019 when FM-A was operated. However, the geolocational patterns for ascending and descending orbits markedly differed from those of the FM-B periods, suggesting that the mechanism introducing the enhanced frequency noise acts differently on the two laser transmitters, potentially due to the different locations in the payload. The underlying reason for the observed dependence on geolocation could be traced back to the reaction wheels of the satellite, as will be explained in the following section.

### 3.3 Influence of the reaction wheels

Precise three-axis attitude control of the Aeolus satellite is accomplished by a set of reaction wheels (RW) which rotate at different speeds, thereby causing the spacecraft to counterrotate proportionately through the conservation of angular momentum. Due to external disturbances, mainly aerodynamic drag, the total angular momentum is periodically modified so that magnetorquers are additionally required to generate an effective external torque. Otherwise the wheel speed would gradually increase in time and reach saturation (Markley and Crassidis, 2014). The attitude and orbit control system of Aeolus additionally consists of thrusters which allow for larger torque to be exerted on the spacecraft.

A sketch illustrating the orientation of the reaction wheels within the spacecraft is shown in Fig. 7. The reaction wheel assemblies (RWAs) are mounted on the spacecraft such that they form a tetrahedron whose axis of symmetry lies along the +Z axis, i.e. the line-of-sight direction of the telescope. Hence, the normal vectors of the four wheels span a plane parallel to the X-Y plane (Fig. 7(b)) in which the two lasers (including the respective MO axes) are located. Each wheel is canted such that the angle between its spin axis and each spacecraft axis is 54.74°. The reaction wheels were manufactured by Stork Product Engineering B.V. in 2005 (design later transferred to Moog Bradford) and have a capacity of 40 Nms and maximum torque of 0.2 Nm. These wheels consist of a rotating inertial mass driven by a brushless DC motor (Bradford space, 2021) and supported by oil-lubricated bearings. The wheels are all mounted on isolation suspensions to reduce micro-vibrations.

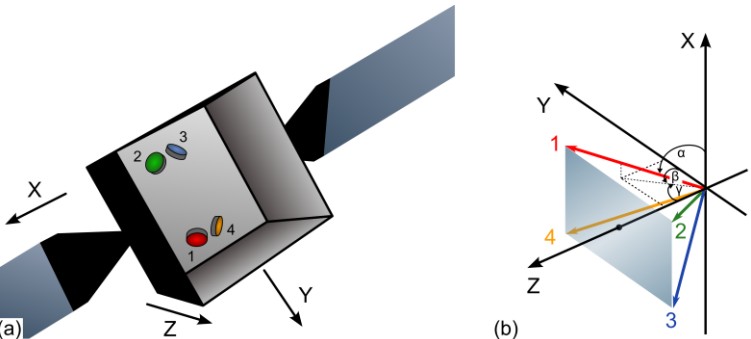

**Figure 7.** (a) Orientation of the four reaction wheels with respect to the spacecraft body. (b) Alignment of the reaction wheel spin axes with respect to the spacecraft body. The axes of the four wheels are oriented such that $|\alpha| = |\beta| = |\gamma| = 54.74°$.

Based on the dataset from the week in October 2019, the frequency stability on observation level, as depicted in Fig. 6, was correlated with the rotational speed of the three active reaction wheels on-board Aeolus (RWA 4 serves as a backup). The same procedure was performed for the FM-A period in May 2019 (see Table 1). The resulting six correlation plots, which can also be considered as spectra in terms of the wheel rotation frequency (rotations per second, RPS), are shown in Fig. 8. Note that the plot includes data from both ascending and descending orbits, and that RWA 1 and RWA 3 rotate counterclockwise, while RWA 2 rotates clockwise. For the sake of better comparability of the three spectra, the absolute wheel speeds are plotted in the figure and the negative sign for the wheel speeds of RWA 1 and RWA 3 was omitted. The six spectra exhibit pronounced peaks which demonstrates that the laser frequency fluctuations are enhanced at specific rotational speeds of the reaction wheels. Thus, the latter are subsequently referred to as critical wheel speeds or critical frequencies.





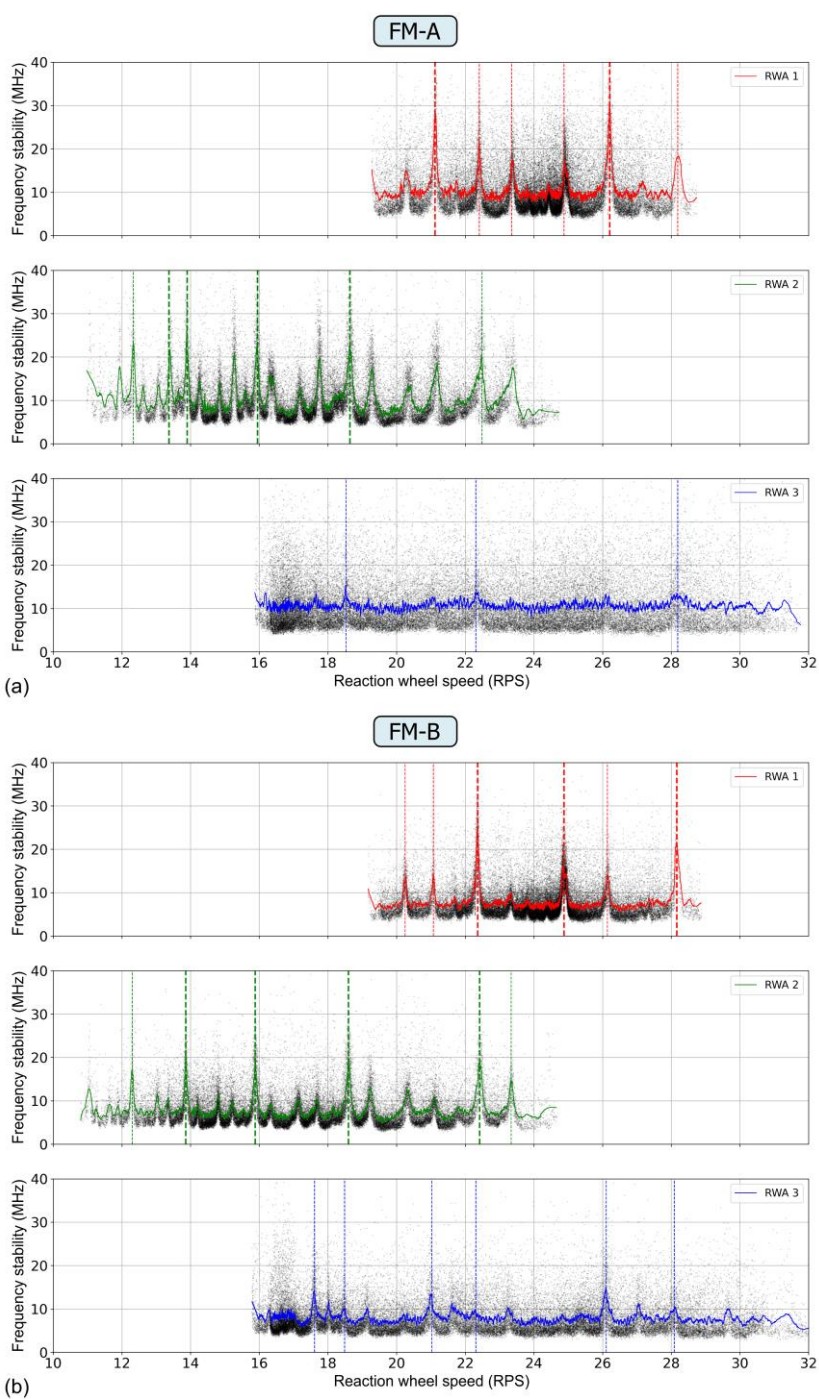

**Figure 8.** Correlation between the laser frequency stability and the speeds of the three active reaction wheels: The plots in panel (a) are based on data from the FM-A laser for the period between 13 May 2019 and 20 May 2019, while the plots in panel (b) show the correlation for the FM-B laser in the period between 14 October 2019 and 21 October 2019. The colored lines result from a Savitzky-Golay smoothing with a window size of 201 and polynomial order of two.





For both periods, i.e. operated lasers, the frequency stability is primarily influenced by RWA 1 and RWA 2, which show a
425 multitude of critical wheel speeds with comparable strength in their common operating range between 19 RPS and 24 RPS. In
contrast, the correlation of the frequency stability with the speed of RWA 3 is rather poor, especially for FM-A, which can be
attributed to being located further away in the instrument. For RWA 1 and RWA 2 a stronger correlation to their nearby FM-A
rather than to FM-B, located on the opposite side of the instrument, is not possible to demonstrate. Since the performances of
FM-A and FM-B are compared for different periods of operation, and these units showed very different behavior in terms of
430 laser performance, degradation and susceptibility to temperature variations, it is difficult to judge if one reaction wheel has a
greater disturbing effect on one or another due to being located closer-by.

The center frequencies and $1/e^2$-widths of the strongest peaks were determined from Gaussian fits and are provided in Tables
2 and 3 for FM-A and FM-B, respectively. From a comparison of the two tables, it can be concluded that the same set of
critical wheel speeds appears for both lasers, however with different relative strength. Furthermore, as becomes also apparent
from Fig. 8, the critical frequencies are consistent among the three wheels. For instance, all wheels feature the same critical
frequency of around (22.4 ±0.1) RPS. Since the three wheels span over different speed ranges, only a subset of critical speeds
is identified for each wheel. By combining the information obtained from all spectra, the following seven reaction wheel speeds
were identified to be most critical regarding the frequency stability of both lasers (sign neglected): $\omega_1 \approx 13.9$ RPS,
$\omega_2 \approx 15.9$ RPS, $\omega_3 \approx 18.6$ RPS, $\omega_4 \approx 22.4$ RPS, $\omega_5 \approx 24.9$ RPS, $\omega_6 \approx 26.2$ RPS and $\omega_7 \approx 28.2$ RPS. These frequencies are
440 printed in bold italics in Tables 2 and 3.

**Table 2.** Critical wheel speeds for the analyzed period between 13 May 2019 and 20 May 2019 (FM-A period), as derived from the
correlation with the frequency stability depicted in Fig. 8(a). The values represent the center frequency and $1/e^2$-width from Gaussian fits
applied to the six most pronounced peaks. The strongest peaks for each reaction wheel are indicated in bold type, while those frequencies
which are critical for both lasers are additionally printed in italics.

| RWA 1 | RWA 2 | RWA 3 |
|---|---|---|
| **(21.12 ± 0.08) RPS** | (12.33 ± 0.08) RPS | |
| *(22.41 ± 0.09) RPS* | **(13.37 ± 0.09) RPS** | (18.53 ± 0.11) RPS |
| (23.35 ± 0.10) RPS | ***(13.90 ± 0.07) RPS*** | |
| *(24.88 ± 0.13) RPS* | ***(15.95 ± 0.08) RPS*** | (22.31 ± 0.10) RPS |
| *(26.21 ± 0.10) RPS* | ***(18.64 ± 0.12) RPS*** | |
| *(28.19 ± 0.13) RPS* | ***(22.49 ± 0.15) RPS*** | (28.19 ± 0.13) RPS |





**Table 3.** Critical wheel speeds for the analyzed period between 14 October 2019 and 21 October 2019 (FM-B period), as derived from the correlation with the frequency stability depicted in Fig. 8(b). The values represent the center frequency and $1/e^2$-width from Gaussian fits applied to the six most pronounced peaks. The strongest peaks for each reaction wheel are indicated in bold type, while those frequencies which are critical for both lasers are additionally printed in italics.

| RWA 1 | RWA 2 | RWA 3 |
|---|---|---|
| (20.25 ± 0.09) RPS | (12.30 ± 0.07) RPS | (17.61 ± 0.10) RPS |
| (21.06 ± 0.08) RPS | *(13.86 ± 0.08) RPS* | (18.49 ± 0.10) RPS |
| *(22.36 ± 0.09) RPS* | *(15.88 ± 0.10) RPS* | (21.03 ± 0.14) RPS |
| *(24.88 ± 0.14) RPS* | *(18.60 ± 0.13) RPS* | (22.31 ± 0.12) RPS |
| *(26.14 ± 0.12) RPS* | *(22.42 ± 0.15) RPS* | *(26.11 ± 0.11) RPS* |
| *(28.16 ± 0.10) RPS* | (23.34 ± 0.11) RPS | (28.09 ± 0.13) RPS |

The variability in the center frequency of the common critical wheel speeds is on the order of 0.1 RPS which is comparable to the average width of the fitted peaks. The fact that the peaks are relatively narrow explains the rather short duration of the high-noise periods of several tens of seconds, as the critical wheel speeds are usually passed on these timescales. Analysis of the other three periods listed in Table 1 yields that the center frequencies and widths of the peaks are constant over time for each laser. Interestingly, the peak heights in the spectra for the early FM-A period in December 2018 are lower compared to the May 2019 period plotted in Fig. 8(a). This is due to the better overall performance of FM-A, particularly the better MO alignment, at the beginning of the mission (see also Table 1), so that the laser was less prone to external perturbations. Consequently, the frequency stability was less degraded at the critical wheel speeds, which manifests in smaller peaks in the spectra.

Steering of the satellite pointing by means of the reaction wheel speeds involves regular and repeated patterns over the one-week orbit repeat cycle, which differ only slightly depending on the elapsed time since the last orbit correction maneuver. The global occurrence of the most critical wheel speeds during the analyzed week in October 2019 is illustrated in Fig. 9. The two maps show those observations during ascending (panel (a)) and descending orbits (panel (b)) for which one of the wheels operates at one of the critical frequencies that are printed in bold type in Table 3. Comparison of the geolocational patterns with those of the laser frequency stability (Fig. 6) underlines the strong correlation between the reaction wheel speeds and the laser frequency noise. For instance, the manifestation of high noise along a linear structure that extends from South and North America across the Pacific Ocean to East Asia for ascending orbits can be traced back to the influence of RWA 1 operating at the critical frequency of $\omega_4 \approx 22.4$ RPS. The enhanced noise which is observed in the north polar regions, on the other hand, is primarily related to RWA 2 rotating at a speed of $\omega_5 \approx 24.9$ RPS. It should be pointed out that the sequence of operating speeds for each wheel slightly changes from week to week depending on the actual orbit position of the Aeolus spacecraft. Consequently, the areas which experience high frequency noise have been shifting by a few hundreds of km over the course of the mission.



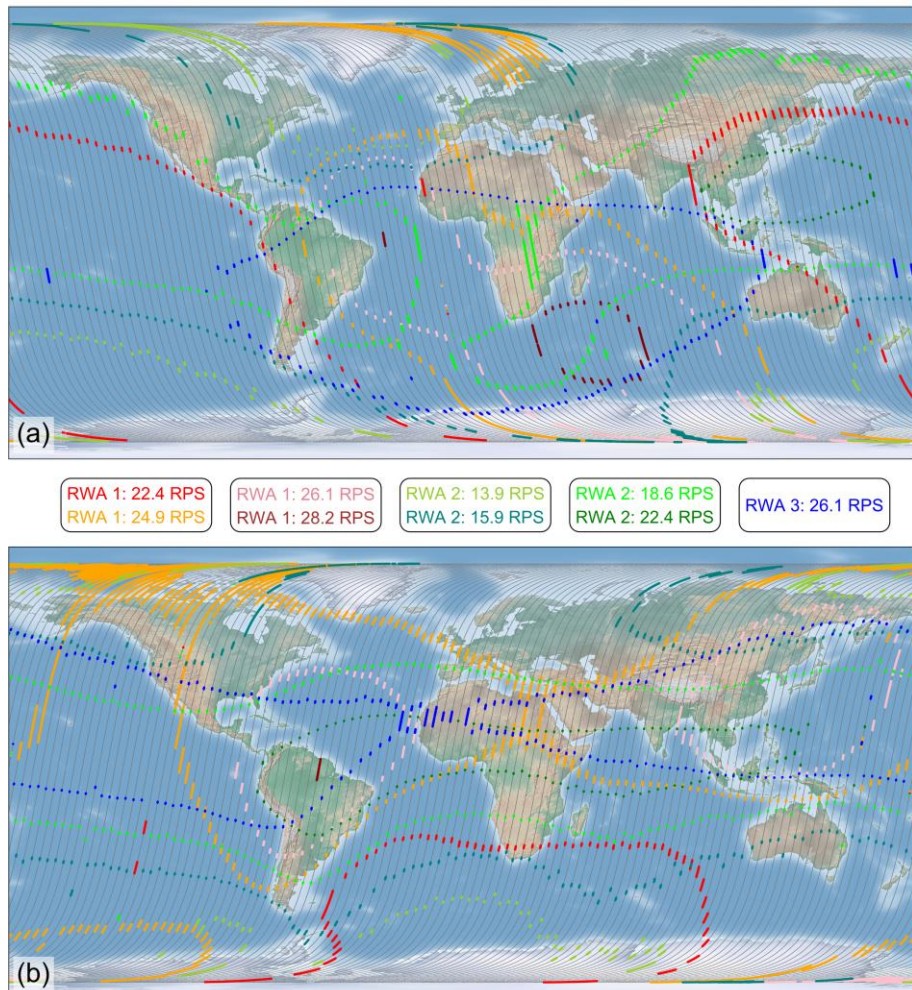

**Figure 9.** Geolocation of the most critical frequencies of the three active reaction wheels for (a) ascending and (b) descending orbits. The plot contains the data from the week between 14/10/2019 (0 UTC) and 21/10/2019 (0 UTC) (see also Fig. 6 for the corresponding frequency stability). Each dot corresponds to one observation (12 s) for which one of the wheels operates at one of the critical wheel speeds as listed in bold type in Table 3.

The occurrence of critical frequencies from different wheels along the orbit suggests that the three wheels act independently on the laser. This hypothesis was confirmed by further analysis which revealed that enhanced noise is observed almost every time when one of the wheels rotates at a critical speed, regardless of the speed of the other two wheels. Hence, there is no entanglement of the critical frequencies, even though the speeds of the wheels are related among each other. Additional studies also showed that the frequency stability is not correlated with the wheel acceleration.

The impact of the reaction wheel speeds on the laser frequency stability is finally demonstrated at an example scene which clearly illustrates the origination of the geolocation patterns. Figure 10 shows a map with the color-coded frequency stability per observation (dots on the map) for the period between 16:30 UTC and 17:30 UTC on 18 October 2019. Within this hour the satellite crossed Africa and Europe on an ascending orbit and passed the North Pole before flying over Alaska and the





Pacific on a descending orbit. The inset of the figure depicts the temporal evolution of the wheel speeds of RWA 1 (red) and RWA 2 (green) which are most relevant regarding the laser frequency stability, as stated above. The timeline of the latter is

plotted below. The critical speeds of RWA 1 and RWA 2 that were identified from the correlation plots in Fig. 8(b) are indicated by horizontal dashed lines of the respective color in the top panel of the inset. The nine marked spots indicate events when the wheels rotated at their critical speeds. As can be seen from the figure, such events are correlated with enhanced frequency variations that result in standard deviations of more than 15 MHz. The frequency noise was especially high in the period between 16:50 UTC and 17:15 UTC when the wheel speed of RWA 1 remained very close to $\omega_5 \approx 24.9$ RPS, hence

leading to a plateau of increased jitter with additional spikes at times when RWA2 rotated at $\omega_1 \approx 13.9$ RPS or $\omega_3 \approx 18.6$ RPS. As shown in this example, the existence of critical wheel speeds explains the enhanced frequency jitter

    a)   on longer periods, i.e. along orbits. That is the case when the speed reaches a local maximum or minimum that lies close to one of the critical values and results in a longer-term perturbation of the laser over several minutes.

    b)   at isolated points in time forming the linear and circular patterns around the globe. That is the case when the speed

quickly crosses a critical value so that only very few consecutive observations are affected. Such events occur at different geolocations for ascending and descending orbits, but are largely reproducible from week to week.

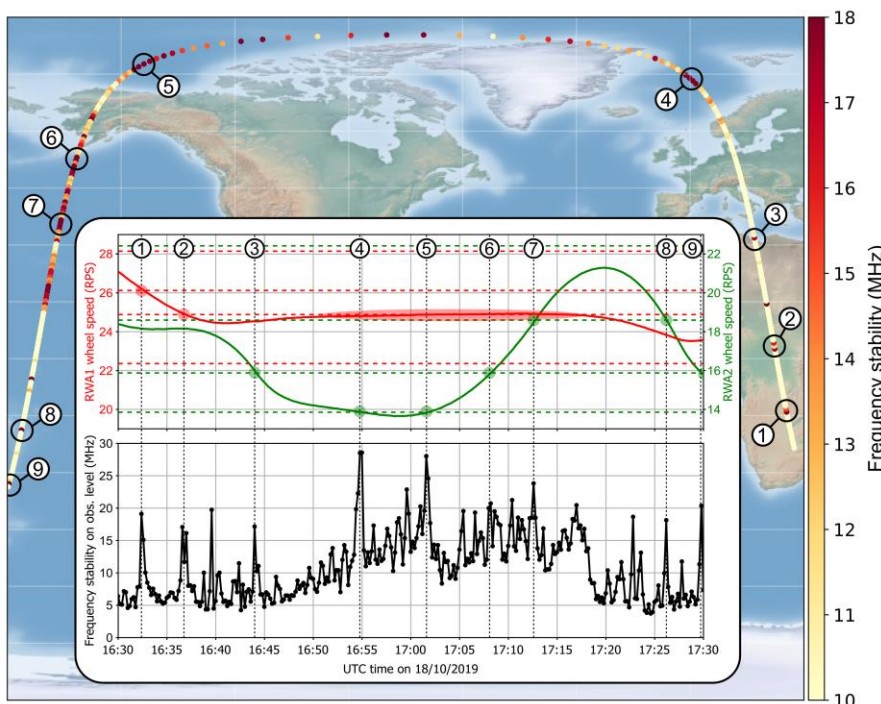

**Figure 10.** Geolocation of observations with enhanced laser frequency variations and their relation with the reaction wheel speeds of RWA 1 and RWA 2 in the period from 16:30 UTC to 17:30 UTC on 18/10/2019. Each dot in the map corresponds to one observation, whereby the

color-coding describes the standard deviation of the relative frequency on pulse-to-pulse level. The inset depicts the temporal evolution of the RWA 1 (red) and RWA 2 (green) wheel speeds on the top panel together with the frequency stability per observation. Horizontal dashed lines in the top panel indicate the most critical wheel speeds that were determined for the analyzed week in October 2019 (see Table 3). Additional peaks in the frequency stability are related to other critical frequencies which are not listed in Table 3 but visible in Fig. 8.





The laser frequency fluctuations for the presented scene are additionally shown on pulse-to-pulse level in Fig. 11(a) together

with the Allan deviation calculated for three selected time series. The Allan deviation (Allan, 1966) is a statistical means to determine the frequency stability over a wide range of averaging times which allows to identify different types of noise sources and drift components which affect the stability on different time scales. While the black data points represent the entire 1-hour period, the green and red data points correspond to 20-minute subperiods with comparatively low and high frequency noise, respectively. The Allan deviation provides further information on the frequency stability on those time scales that are relevant

for the wind retrieval. Apart from the observation level (12 s, 540 pulses) the frequency variations on measurement level (0.4 s, 18 pulses, see section 2.1) are crucial for the wind data quality. This is due to the fact that the signal data from individual measurements is classified into "clear" and "cloudy" bins by using estimates of the backscatter ratio before the on-ground accumulation to so-called groups in the L2B processor (Rennie et al., 2020). This grouping algorithm allows to distinguish between measurements that are better analyzed with the Rayleigh or Mie spectrometer, respectively, and hence to minimize

the effect of crosstalk between the two receiver channels that is detrimental to the wind data quality. In particular, Mie wind results are usually provided on time scales shorter than 12 s, as they require fewer measurement bins to achieve a given level of precision compared to the Rayleigh winds, with the typical levels of backscatter, e.g. from clouds (Rennie et al., 2020).

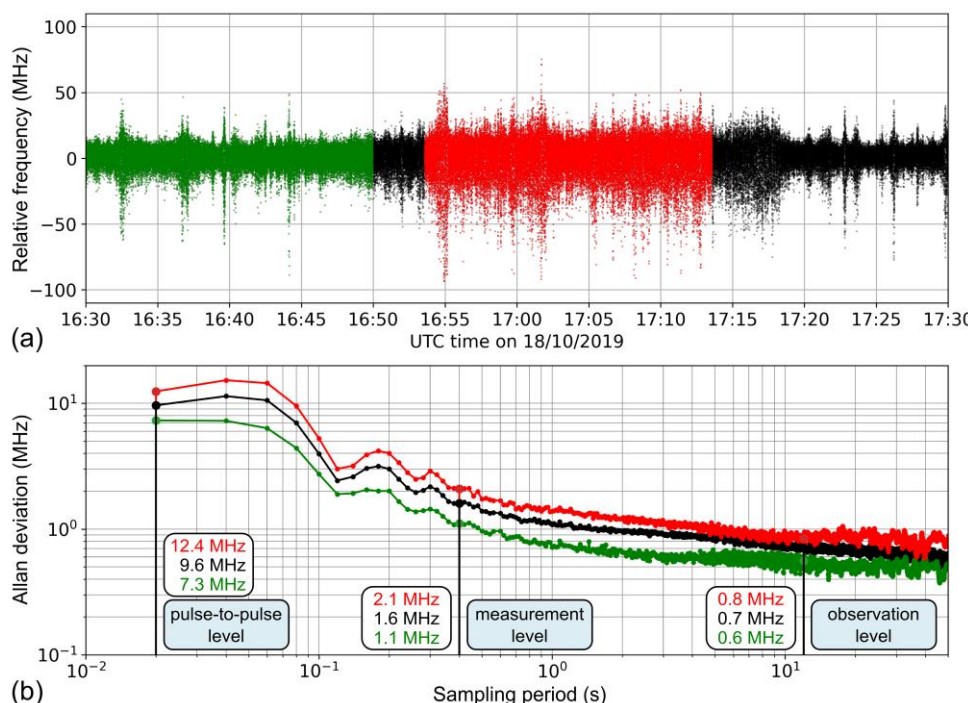

**Figure 11.** Laser frequency stability on pulse-to-pulse level for the period from 16:30 UTC to 17:30 UTC on 18/10/2019 (see also Fig. 10). (a) Time series of the relative frequency. From the overall 1-hour period, two intervals of 20 minutes with good (green) and poor frequency stability (red) were selected for further analysis. The bottom plot represents the Allan deviation calculated for the selected time series in the upper panel. The vertical dashed lines indicate sampling periods relevant for the wind retrieval (pulse-to-pulse level – 0.02 s, measurement level – 0.4 s, observation level – 12 s).




530 For the low-noise period, the Allan deviation on measurement level (0.4 s sampling period) is 1.1 MHz which is about 50% better than for the entire 1-hour period (1.6 MHz) and almost two times better compared to the high-noise period (2.1 MHz). The differences are less pronounced when sampling on pulse-to-pulse or observation levels is regarded. This underlines that the enhanced frequency fluctuations typically occur in the second regime (see also Fig. 3(b)), so that the mean laser frequency varies strongly from one measurement to the other. On longer time scales, the enhanced noise has a minor impact on the mean

535 frequency, i.e. the variation is smaller from observation to observation. As a result, the Allan deviation on observation level is around $(0.7 \pm 0.1)$ MHz almost independent on the occurrence of enhanced noise periods.

Although the root cause of the enhanced frequency noise is understood to be introduced by the reaction wheels, there is an apparent correlation with other platform parameters. In particular, a link to the data obtained from the magnetometer on-board Aeolus was discovered. A thorough investigation revealed a strong influence from the magnetic fields generated by the

540 platform magnetorquers which serve the regulation of the reaction wheel speeds. Due to the complex coupling between magnetorquer currents and reaction wheel speeds, there is an indirect relationship between the critical wheel speeds, i.e. the frequency stability, and the magnetometer data via the superimposed magnetic fields of the magnetorquers and the Earth.

### 3.4 Micro-vibrations as the root cause for increased frequency jitter

The observation of critical speeds of the reaction wheels that give rise to enhanced frequency noise strongly suggests that

545 micro-vibrations are the root cause of the degraded laser frequency stability in orbit. This hypothesis is further strengthened by the fact that the frequency noise levels also increase during periods of heavy thruster firings, e.g. during orbit maneuvers and wheel speed set point changes before and after IRCs (see section 3.2).

When the thrusters are not used, the dominating micro-vibrations occurring in the satellite are a result of its structure responding to disturbing forces and moments generated by the reaction wheels. These disturbances result mainly from static

550 and dynamic unbalances of the flying wheel, bearing imperfections and the structural modes of the wheel assembly, and are exerted on the satellite structure as a combination of harmonics, each with a frequency being a constant multiple of the wheel speed. Typically, these disturbances become the most intense at the wheel speeds at which the critical harmonics excite the structural modes of the wheel assembly. Please refer to Le (2017) for a detailed characterization and study of the mechanical disturbances generated by reaction wheels of the class of those embarked on Aeolus.

555 For Aeolus the reaction wheels were mounted on micro-vibration isolation suspensions, which filter the generated disturbance forces and moments. Such suspension was found to reduce the disturbances by more than one order of magnitude at the most critical frequencies, while the amplifications occurring at the lower frequency due to the suspension modes are minimized by the presence of viscoelastic elastomeric mounts.

The on-ground micro-vibration verification activities were quite extensive within the Aeolus project. These included disturbing

560 the laser transmitter with representative mechanical excitation spectra, which allowed identifying particularly susceptible in the 400 Hz to 600 Hz frequency band, as well as around 250 Hz. Moreover, micro-vibration tests were performed at satellite-level. First, with the aim of characterizing the micro-vibration environment throughout the satellite, which was achieved by




operating the reaction wheels all over their operational speed range, and having the satellite mounted on doughnut-shaped cushions for isolating it from external disturbances (Lecrenier et al., 2015). These tests demonstrated that, despite the fact that the peak of disturbances from the reaction wheels coincided with the most susceptible frequencies of the lasers, the vibration levels were lower than the danger-levels previously established to them, thanks to the isolation suspension. The margins observed between the micro-vibration levels measured during the full-satellite test and the danger-levels characterized during the preceding laser tests were considered large enough to cover for the mechanical changes that occur when passing from the ground to the orbital environment. Due to the different support boundary conditions of the satellite and the effects of gravity and air, the structural damping and natural frequencies of the satellite structure are expected to differ on-orbit from their characterization on-ground. Moreover, a worsening of the disturbance signature of the reaction wheels as a result of their exposure to the vibrations of the satellite-level tests and the launch environment is also commonly observed.

These margins were finally confirmed by an in-situ test during the thermal vacuum campaign with the flight model of the spacecraft. This included operation of the laser whilst the wheels were running at speeds previously identified as critical and after wheel power off. The Mie and Rayleigh frequency response data from these tests clearly showed significant peaks at several harmonic frequencies (roughly 4.5, 10, 13, 16, 18.5, 21.5, 23, 24, 24.5 RPS) when the wheels were operating. In fact, the shot-to-shot performance varied along the thermal plateau stronger than during the dedicated reaction wheel test. Unfortunately, due to programmatic constraints, no tests were ever run with the spacecraft mounted in an isolated configuration from the ground and with flight-representative time-varying speed profiles whilst operating the laser to check its frequency stability.

## 4 Impact of the frequency fluctuations on the data quality

After having identified the root cause of the temporally degraded frequency stability in orbit, the question is to what extent the Aeolus wind data quality is diminished during the periods of enhanced frequency noise. Before answering this question, it is meaningful to consider how often these periods occur and how long they last. For this purpose, the percentage of observations for which the frequency stability is significantly worse in comparison to the overall performance over a certain period can be regarded. This information is summarized in Table 4 for the five selected weeks of the mission that were already introduced in section 2.4 as being representative for different periods of the Aeolus mission.





**Table 4.** Laser frequency stability at different phases of the Aeolus mission. The table provides the standard deviation of the relative frequency on pulse-to-pulse level (mean over all observations from one week) as well as the percentage of observations for which the standard deviation is above 15 MHz, 20 MHz and 25 MHz, respectively. FM-A was operating during the first two selected periods, while FM-B was operating in the other three weeks. See also the time series of the frequency stability for the week in October 2019 in Fig. 4.

| Period | σ over entire week | %obs with σ > 15 MHz | %obs with σ > 20 MHz | %obs with σ > 25 MHz |
|---|---|---|---|---|
| 17/12/2018 - 24/12/2018 (FM-A) | 7.2 MHz | 4.6% | 0.9% | 0.2% |
| 13/05/2019 - 20/05/2019 (FM-A) | 10.7 MHz | 18.2% | 7.7% | 3.0% |
| 14/10/2019 - 21/10/2019 (FM-B) | 8.1 MHz | 7.2% | 2.4% | 0.7% |
| 17/08/2020 - 24/08/2020 (FM-B) | 8.7 MHz | 8.9% | 3.5% | 1.3% |
| 28/09/2020 - 05/10/2020 (FM-B) | 8.6 MHz | 7.8% | 2.6% | 0.7% |

The table provides the standard deviation over the entire week and the portion of observations for which the standard deviation is above 15 MHz, 20 MHz and 25 MHz. The data for the week in October 2019 is also visualized in Fig. 4(a). The worst performance was evident at the end of the FM-A period in May 2019 when the MO was already strongly misaligned. Consequently, the frequency stability calculated over the whole week was 10.7 MHz, while nearly one fifth of all observations showed frequency variation larger than 15 MHz. This corresponds to an increase by a factor of 4 compared to the early FM-A phase in December 2018 when less than 5% of all observations were affected by enhanced noise of that order. Regarding extreme cases, i.e. when the standard deviation exceeded 25 MHz, the percentage has even multiplied by 15 from 0.2% to 3.0% over the course of the FM-A period. On the contrary, the performance of FM-B is rather stable with the frequency stability ranging between 8 and 9 MHz and about 1% of observations with σ > 25 MHz. As of October 2020, nearly 8% of all observations showed frequency fluctuations above 15 MHz and for less than 3% the standard deviation was above 20 MHz.

In the following two sections, the influence of the frequency stability on the systematic and random error will be assessed for the Mie and Rayleigh wind results as well as for the respective ground velocities. While the latter study is performed on observation level (540 pulses), the impact on the wind results is discussed on measurement level (18 pulses) for the reasons related to the wind retrieval processing that were explained at the end of section 3.3.

## 4.1 Accuracy and precision of the Aeolus Rayleigh and Mie winds

The wind error assessment is based on the data from two weeks of FM-B operation in August and September/October 2020, as listed in Tables 1 and 4. These datasets already include the correction for the influence of the temperature variations across the primary telescope mirror on the wind results (M1 correction) which was implemented prior to the operational assimilation of the Aeolus wind data in NWP by various weather services (Rennie and Isaksen, 2020a). The Rayleigh and Mie wind observations were extracted from the L2C product which, in addition to a copy of the Aeolus L2B product, includes ECMWF model winds (analysis and background) provided on the same horizontal and vertical grid. Using the background model winds as the reference, the wind speed differences between observation and background (O-B) can be interpreted as the wind errors.





The L2B/C wind data is provided on a different temporal grid as the L1A data which is used for the calculation of the Mie responses and corresponding frequency fluctuations. This is due to the classification of measurement bins into "clear" and "cloudy" bins (section 3.3) which results in so-called groups of varying horizontal length. Consequently, for investigating the influence of the frequency stability on the wind accuracy and precision, an adaptation of the different temporal grids of the

620 L1A/B and L2B/C products has to be performed. Moreover, it has to be considered that each wind profile measured within a certain period of time generally comprises multiple wind results from the adjacent vertical bins to be compared with the respective frequency stability within the regarded time interval. Finally, only reliable wind results with low estimated error should enter the statistics. The estimated wind error is included in the L2B/C product and, in case of the Rayleigh channel, is derived from the SNR and the pressure and temperature sensitivity of the Rayleigh responses (Rennie et al., 2020). For the

625 Mie channel, it is primarily linked to the SNR. The estimated error also considers the impact of the solar background on the wind accuracy, which is mainly relevant for the Rayleigh winds. In the analysis presented here, estimated error thresholds of 8 m·s$^{-1}$ for the Rayleigh and 4 m·s$^{-1}$ Mie channel were used, as airborne validation campaigns have demonstrated that these values ensure small departures of the Aeolus winds from high-accuracy wind lidar measurements (Witschas et al., 2020).

The results of the statistical analysis for one of the two studied periods is shown in Fig. 12. Here, panel (a) depicts a histogram

of the laser frequency stability on measurement level, i.e. calculated as the standard deviation over 18 pulses within 0.4 s. After interpolation of the frequency stability data onto the temporal grid of the Mie wind data, the former could be filtered for those measurements within the week from 17 to 24 August 2020 that yielded valid Mie winds. From the nearly 350,000 Mie ("cloudy") wind results, more than 78% were obtained at a frequency stability of better than 10 MHz on measurement level, as indicated by the light blue area in the histogram. Regarding the Rayleigh ("clear") winds, the percentage is identical,

although the amount of wind data is about 2.8 times larger (≈990,000 winds) compared to the Mie channel due to its usually larger data coverage. The median of the laser frequency stability is 6.6 MHz for the two data subsets, while the mean values both account for 8.2 MHz which is comparable to the frequency stability on observation level for the August 2020 period stated in Table 4. Although the frequency stability is comparable for the two different processor time scales when the mean over large datasets is considered, the fluctuations are significantly higher on measurement level (>30 MHz) than on observation

level in rare cases when the peak of a short high-noise period falls within one measurement.

In a next step, the Mie and Rayleigh wind results were separated into those for which the frequency stability was better or worse than 10 MHz on measurement level. The resulting probability density functions (PDFs) of the (O-B) horizontal line-of-sight (HLOS) wind speed differences are provided in the middle and right columns of Fig. 12. Due to the much smaller amount of wind results for σ > 10 MHz (≈22%), the distribution is a bit noisier than for σ < 10 MHz. Nevertheless, nearly Gaussian

distributions are evident for the two subsets, both for the Mie and Rayleigh channel. The mean of the Mie distribution, or bias, of the "high-noise" case is slightly larger [(0.214 ± 0.015) m·s$^{-1}$] than that of the "low-noise" case [(0.030 ± 0.008) m·s$^{-1}$]. Hence, when only considering winds from the week in August 2020 for which the frequency stability was worse than 10 MHz on measurement level, the bias differs by about 0.18 m·s$^{-1}$ from the mean (O-B) wind difference over all winds for which σ < 10 MHz. When the wind data is further restricted to σ > 20 MHz, the change in bias increases to 0.31 m·s$^{-1}$ (not shown).





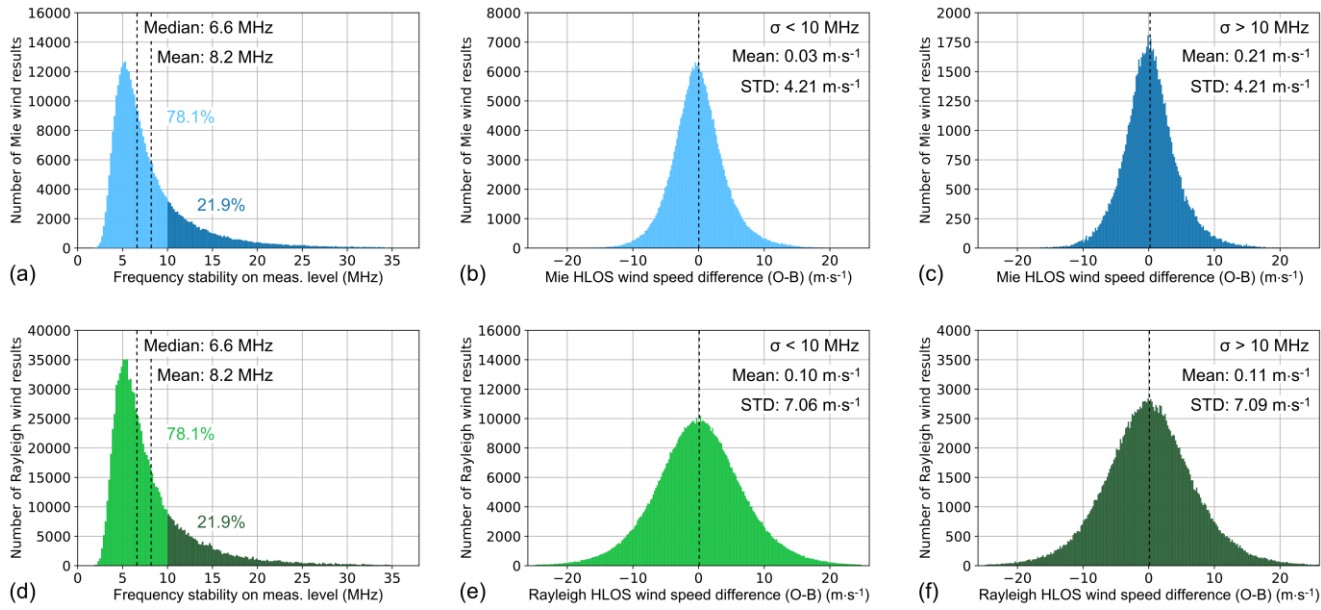

**Figure 12.** (a) Histogram of the laser frequency stability on measurement level for those measurements that yielded valid Mie winds during the period from 17 to 24 August 2020 (bin size: 0.2 MHz). The probability density functions of the wind speed difference with respect to the ECMWF model background (O-B) is shown in panels (b) and (c) for measurements with a frequency stability better and worse than 10 MHz, respectively (bin size: 0.2 m·s$^{-1}$). The bottom panels depict the corresponding statistics for the Rayleigh wind data.

The widths of the two distributions, however, are identical (standard deviation: 4.21 m·s$^{-1}$), showing that the frequency stability has no significant influence on the random error. The same is true for the Rayleigh winds whose PDFs are depicted in Fig. 12(e) and (f). Here, the differences in the bias and random error for σ < 10 MHz and σ > 10 MHz are below 0.04 m·s$^{-1}$. Thus, it can be concluded that the accuracy of the Rayleigh winds is less affected by the enhanced frequency noise than that of the Mie winds, and that the influence on the random wind error is negligible for both channels.

The same analysis was performed for the week between 28 September and 5 October 2020 and yielded similar results. Here, the percentage of Mie and Rayleigh wind results for which the frequency stability on measurement level is better than 10 MHz is 78.6% and 78.9%, respectively. The Mie bias increases from (0.079 ± 0.008) m·s$^{-1}$ for σ < 10 MHz to (0.187 ± 0.016) m·s$^{-1}$ for σ > 10 MHz, while the Rayleigh bias differs by only 0.02 m·s$^{-1}$. The respective random errors are 4.22 m·s$^{-1}$ and 4.28 m·s$^{-1}$ for the Mie and 7.16 m·s$^{-1}$ and 7.17 m·s$^{-1}$ for the Rayleigh channel, respectively.

In order to further evaluate the impact of the laser frequency noise on the Aeolus wind data quality, a variable frequency stability threshold was applied and the bias and random error with respect to the ECMWF model background were calculated for those winds that were measured during periods with frequency fluctuations below the threshold. The statistical results are depicted in Fig. 13. The mean Mie and Rayleigh wind bias are plotted in the left column, while the respective random errors for the two investigated weeks in August and September/October 2020 are shown on the right. Both values are presented relative to the statistical parameters that are obtained for a threshold of 8 MHz, as this allows for a direct comparison of the two different datasets regardless of the absolute values which differ among the analyzed periods and receiver channels due to





other error contributions. The reference bias and random error values are indicated in the round boxes. A frequency stability
of 8 MHz was chosen as a reference threshold, since this value is close to the mean stability over the entire week and thus
represents the average conditions in terms of laser frequency noise. The plots additionally present the number of winds that
entered the statistics depending on the applied threshold. As can be seen from the figure, the number of wind results increases
considerably when the threshold is relaxed from 8 MHz to 15 MHz, whereas only few results are added at even higher
thresholds. Overall, the amount of Rayleigh (≈1,000,000) and Mie winds (≈350,000) is comparable for the two periods.
Regarding the bias, a noticeable impact of the frequency stability is only evident for the Mie winds, as seen before. Here, the
bias increases by up to 0.05 m·s$^{-1}$ when wind data from high-noise periods is included in the statistics. For the Rayleigh winds,
the bias change is negligible (<0.01 m·s$^{-1}$). The same holds true for the random errors of both channels which change by less
than 0.02 m·s$^{-1}$ when the frequency stability threshold is relaxed from 8 MHz to more than 20 MHz.

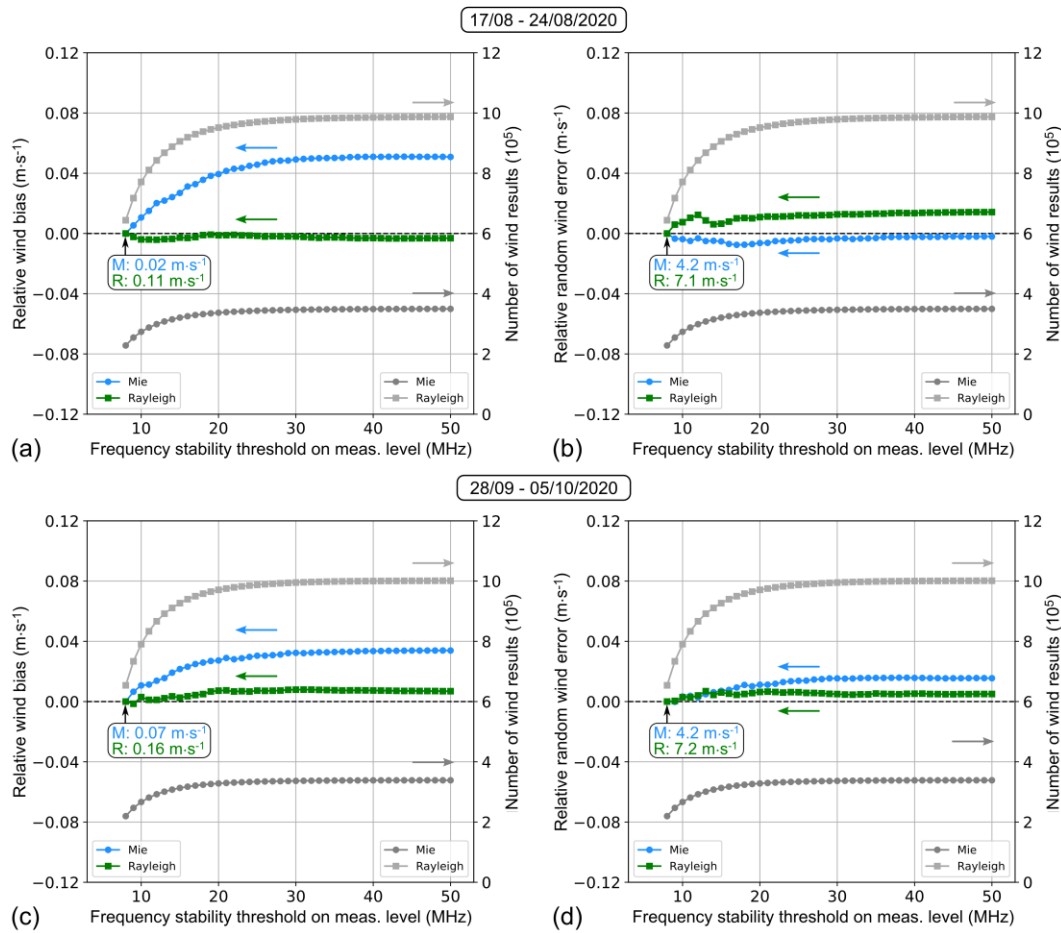

**Figure 13.** Wind bias (a) and random error (b) of the Mie (blue dots) and the Rayleigh channel (green squares) with respect to the ECMWF
model background (O-B) depending on a frequency stability threshold for the period between 17 and 24 August 2020. The bias and random
errors were subtracted by the respective values obtained for a threshold of 8 MHz for the sake of comparability (see reference values in the
boxes). The number of Mie and Rayleigh wind results for which the frequency stability is below the threshold are shown in light grey dots
and dark grey squares, respectively. Panels (c) and (d) show the corresponding data for the week between 28 September and 5 October 2020.





In conclusion, the temporally degraded frequency stability of the ALADIN laser transmitter has only minor influence on the wind data quality on a global scale. This is primarily due to the small percentage of measurements for which the frequency

fluctuations are considerably enhanced. The biggest impact is observed for the Mie wind bias, which is increased by more than 0.3 m·s⁻¹ when considering only those phases with strong variations (>20 MHz) on measurement level. However, as these winds represent only about 4% of all wind results from the studied weeks, the effect is hardly noticeable in the statistics derived from the entire global dataset. Nevertheless, this study helps to identify an optimal threshold for a quality control (QC) that filters out measurements for which the frequency noise is considerably enhanced. The according QC parameter which describes

the frequency stability in terms of the standard deviation on measurement level is already included in the L1B product.

Since wind observations with enhanced frequency jitter occur over specific geolocations (see Fig. 6), it is interesting to study whether the wind data is significantly degraded in such affected locations. For this purpose, several areas where one or multiple critical reaction wheel speeds regularly occur, i.e. in East Asia or Central Africa, were analyzed with regards to the deviations from Aeolus observations from ECMWF model winds (O-B). This preliminary study revealed that, although the percentage

of measurements with frequency stability worse than 15 MHz is increased by a factor of two to three compared to the global percentage, the change in wind bias and random error is below 0.5 m·s⁻¹. However, as the number of wind results is drastically decreased by the restriction to rather small geographical boxes, these results lack statistical significance. Hence, additional studies using longer periods of wind data are required to verify this small influence on the wind data quality on a local scale. Furthermore, as outlined in the end of section 2.1, the impact of enhanced frequency noise is most pronounced in cases when

only a small subset of the emitted pulses is detected in the atmospheric path, i.e. in heterogenous atmospheric scenes (Marksteiner et al., 2015). Therefore, proper extraction of such scenes, e.g. broken cloud conditions, is advisable prior to the statistical analysis. Nonetheless, the current study has provided a strong hint that the laser frequency stability is not a major contributor to the systematic and random errors of the Aeolus wind product for the SNR cases analyzed.

## 4.2 Influence on the Rayleigh and Mie ground velocities

In analogy to the approach presented in the previous section, the correlation between the laser frequency stability and the apparent velocity of the ground returns was investigated. Ground return signals are generally crucial for airborne and spaceborne radar and lidar systems that rely on the Doppler effect, as they can be exploited for identifying systematic errors that are, for instance, caused by improper knowledge of the platform attitude or variations in the instrument's alignment (Bosart et al., 2002; Kavaya et al., 2014; Chouza et al., 2016). For the Aeolus mission, the ground surface could be used as a zero-

wind reference, which allows to estimate unknown wind biases from the measured ground velocities. This method, however, requires precise differentiation between atmospheric and ground return signals in order to prevent erroneous ground velocities (also referred to as Zero Wind Calibration or ZWC values), which is particularly challenging due to ALADIN's coarse vertical resolution of several hundred meters. The ground detection scheme and its limitations are very similar for ALADIN and the A2D and are explained in detail in Weiler (2017) and Lux et al. (2018).





The ground velocities, obtained separately for the Rayleigh and Mie channel, are contained in the AUX_ZWC product which is generated by the Aeolus L1B processor. The same two weeks in 2020 as in the previous section were studied. However, in contrast to the wind results, the ground velocities which are part of the L1B baseline 10 data, are not bias corrected using the M1 telescope mirror temperatures as for the L2B product so that they show rather large deviations from the "ideal value" of 0 m·s$^{-1}$ (Rayleigh: –18 m·s$^{-1}$, Mie: –2 m·s$^{-1}$). Hence, the analysis relies on relative changes in the ground velocities depending

on the frequency stability, while the influence of attitude control mis-pointing on this parameter can be neglected. Moreover, in contrast to the multitude of Rayleigh and Mie wind results which are available from multiple range gates and all over the globe, ground return signals that are strong enough to be utilized for a potential ZWC of the Aeolus data are rare. Given the susceptibility to atmospheric contamination, only ground velocities that were measured from surfaces with high albedo in the UV spectral range are considered valid, which drastically limits the number of available ZWC values. The useful signal

thresholds applied in the L1B processor are 1000 ACCD counts and 1200 ACCD counts for the Mie and Rayleigh channel, respectively. For this reason, the ZWC data from the two weeks were combined in order to obtain meaningful statistics. Since only data from high-albedo regions, i.e. Antarctica, Arctic, sea ice as well as snow covered land areas, entered the statistics, solely the frequency stability over these locations plays a role in the analysis. Since the ice coverage was comparable in the two weeks in 2020, the studied locations and thus the influence of the wheel speeds on the respective frequency fluctuations

was similar for the individual periods that contributed nearly the same amount of ground velocity values (≈4000) to the combined dataset. Note also that the ZWC data is provided only on observation level (12 s = 540 pulses), as there is no differentiation between "clear" and cloudy" bins. Consequently, the ground velocities are correlated with the laser frequency variations on observation level.

In analogy to Fig. 12(a) and (d), histograms of the laser frequency stability on observation level are depicted in Fig. 14(a) and

(c) for those observations that yielded valid Mie and Rayleigh ground velocities, respectively. The number of ZWC values is very similar for the two channels (≈4000), suggesting that almost the same observations, i.e. geographical locations, were considered in the statistical analysis. This also explains the agreement of the median and mean frequency stability on observation level which account for 7.3 MHz and 8.7 MHz, respectively. The fact that these values are slightly higher than those on measurement level presented in the previous section is most likely due to the restriction to polar regions where critical

wheel speeds are overrepresented (see Fig. 9). The influence of the frequency noise on the Mie and Rayleigh ground velocities is displayed in Fig. 14(b) and (d) where the respective mean ground velocities are plotted against the frequency stability which was subdivided into bins of 1 MHz. Here, a data point plotted versus a frequency stability of $k$ MHz represents the mean over those ground velocities that fall in the interval between $(k-1)$ MHz and $k$ MHz. The velocities were subtracted by the mean over all ZWC values for which the frequency stability is better than 8 MHz to exclude the large bias contributions that are

different for the two receiver channels. The error bar represents the corresponding standard deviation per bin. In addition, the number of ground velocities per interval that entered the statistics is plotted in grey.



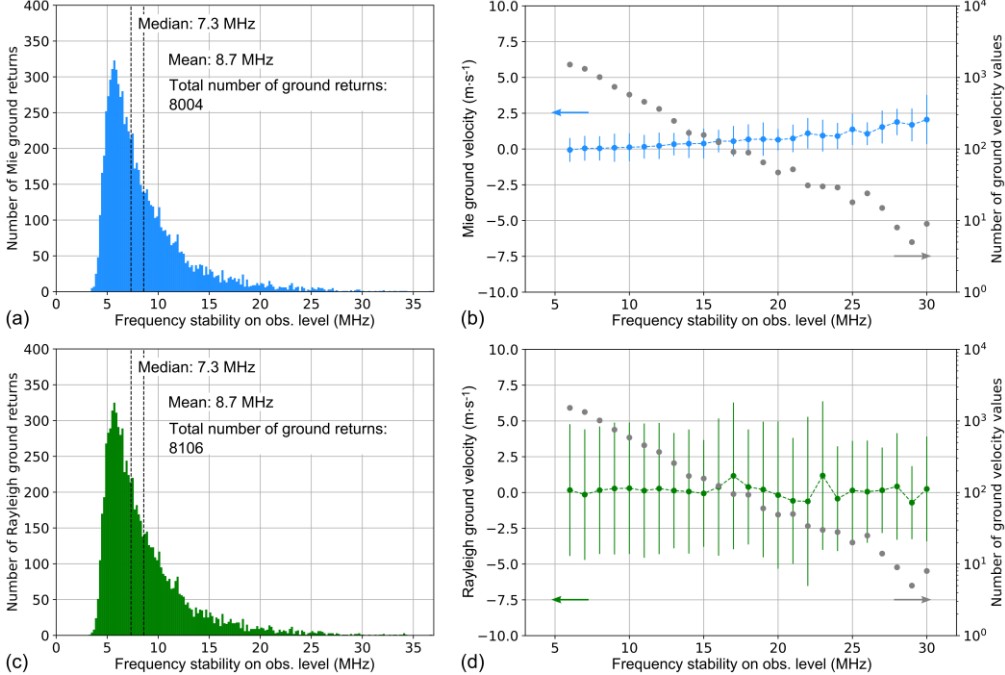

**Figure 14.** Histograms of the laser frequency stability on observation level for those observations that yielded valid Mie ground returns during the combined periods from 17 to 24 August 2020 and from 28 September to 5 October 2020. (b) Mean Mie ground velocity in dependence on the frequency stability. The ground velocities were subtracted by the mean over all values for which the frequency stability is better than 8 MHz for the sake of comparability. A data point plotted versus a frequency stability of $k$ represents the mean over all ground velocities for which the stability is in the interval between $(k − 1)$ MHz and $k$ MHz, while the error bar represents the corresponding standard deviation. The number of ground velocity values within each interval is shown in grey. The bottom panels (c) and (d) depict the corresponding statistics for the Rayleigh ground returns.

Like for the wind results, the number of ZWC values strongly decreases when considering only those observations for which the frequency stability is considerably degraded. The portion of ground velocities with frequency stability worse than 15 MHz is as low as 8.2% for both channels. In accordance with the findings in section 4.1, there is no significant influence of the enhanced frequency noise on the Rayleigh channel, whereas the determined Mie ground velocities change almost linearly with the frequency stability. When regarding only observations with 29 MHz $<\sigma < 30$ MHz (rightmost dot in Fig. 14(b)), the ground velocity is about 2.1 m·s$^{-1}$ higher than for those observations with 5 MHz $<\sigma < 6$ MHz. However, it should also be noted that only nine ZWC values contributed to this last data point, resulting in large uncertainty (standard deviation: 1.7 m·s$^{-1}$) and thus small statistical significance.

The mean ZWC values are additionally calculated for a varying frequency stability threshold, in analogy to Fig. 13, in order to consider the different weighting of the observations with low and high frequency noise. The results are plotted in Fig. 15 for the Mie and Rayleigh channel, both for the mean and the standard deviation of the ground velocities of the respective data subset after applying the threshold. Here, a similarly weak dependency is evident for the Mie and Rayleigh channels where the mean ZWC values (relative to the 8 MHz threshold) increase to 0.15 m·s$^{-1}$ as the threshold is relaxed to 30 MHz. The random errors are even less affected and change by less than 0.10 m·s$^{-1}$ depending on the applied frequency stability threshold.





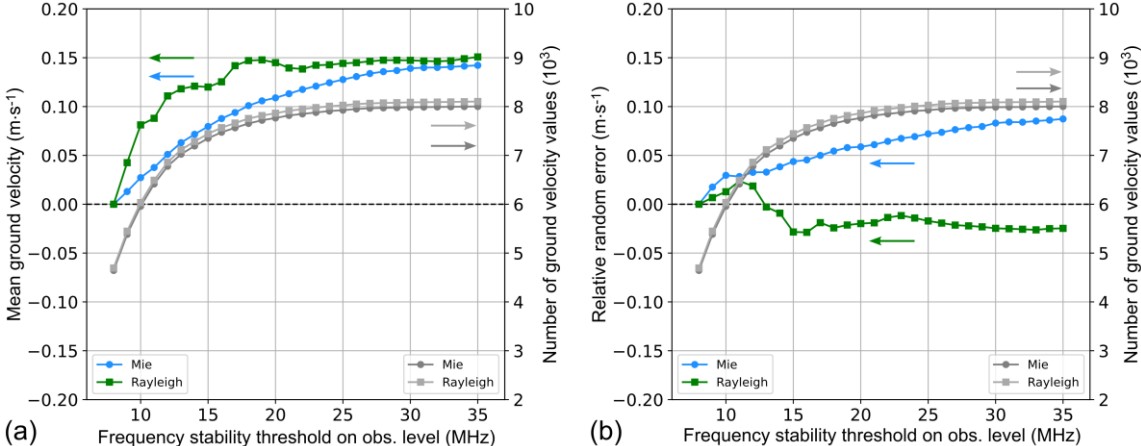

**Figure 15.** Mean ground velocity (a) and relative random error (b) of the Mie (blue dots) and Rayleigh channel (green squares) depending on a frequency stability threshold for the combined periods from 17 to 24 August 2020 and from 28 September to 5 October 2020. The data were subtracted by the respective values obtained for a threshold of 8 MHz for the sake of comparability. The number of ground velocity values for which the frequency stability is below the threshold are shown in grey.

From these results it can be concluded that, like for the atmospheric winds, the enhanced frequency noise has only minor

influence on the Mie and Rayleigh ground velocities. Although the impact is slightly larger, it is still hardly noticeable in the statistics derived from the complete ZWC dataset. Nevertheless, the approximately linear relationship between the frequency stability and the Mie mean ZWC value with a slope of about $0.08 \text{ m·s}^{-1}\text{·MHz}^{-1}$ (Fig. 14(b)) demonstrates that a more severe degradation of the stability would lead to a significant change in the Mie ground velocities. Hence, assuming frequency fluctuations of 50 MHz on observation level, the corresponding Mie ZWC values would deviate by more than $3 \text{ m·s}^{-1}$.

Depending on the contribution of those high-noise observations to the entire dataset, the frequency noise can, in principle, drastically deteriorate the ZWC accuracy. This is especially true if there are only a few valid measurements per observation in case of broken ground or cloud detection. Therefore, application of an appropriate QC threshold, as stated in the previous section, is recommended for the processing of ground velocities as well.

## 5 Summary, conclusions and outlook

The Doppler wind lidar ALADIN on-board Aeolus has set new technological standards in the field of space-borne remote sensing. In particular, the design and performance of the frequency-stabilized UV laser transmitter is unprecedented for a space laser. By the end of 2020, the two ALADIN lasers have accumulated around 3.5 billion high-energy laser pulses in 27 months of operations. The present study has shown that the frequency stability of the emitted pulses is better than 10 MHz, except for the late FM-A period in spring 2019, most probably because of the advanced misalignment of its master oscillator. The

excellent stability is achieved despite the permanent occurrence of short periods during which the frequency fluctuations are considerably enhanced to 30 MHz RMS on observation level (12 s), 50 MHz RMS on measurement level (0.4 s) and even pulse-to-pulse variations of up to 150 MHz peak-to-peak.



The investigation of the frequency stability during instrument response calibrations has revealed that it depends on the geolocation of the satellite. Consequently, the correlation between frequency stability and geolocation could be traced back in
a clustering of observations with enhanced frequency variations in specific regions of the Earth, forming linear and circular structures around the globe. The patterns differ for ascending and descending orbits and the two flight-model lasers, but are stable over the mission life-time.

The underlying reason for the dependency on geolocation is the existence of critical rotation speeds of the satellite's reaction wheels, which suggests that micro-vibrations are the root cause for the deterioration of the laser stability on time scales of a
few tens of seconds. This hypothesis is supported by the fact that the laser stability is also degraded during thruster firings of the satellite which introduce high vibration levels also at the location of the power laser head. The identified detrimental frequencies of the reaction wheels range between 14 RPS and 28 RPS and are consistent among the three active wheels, although the relative impact on the two lasers is different. Owing to the dependency of the reaction wheel speeds on the magnetorquer control authority and the strong influence of the latter on the magnetometer readings, there is an indirect link,
and hence decent correlation, between the frequency stability and the magnetic field measured by the on-board magnetometer. In the context of the enhanced frequency noise, the Aeolus wind error with respect to ECMWF model background winds was studied, pointing out that the temporally degraded stability of the ALADIN laser transmitter has only minor influence on the global wind data quality. For two studied weeks in 2020, the Mie wind bias is increased by 0.3 m·s$^{-1}$ during phases with strong variations (> 20 MHz) on measurement level. Considering the small portion of such wind measurements (≈4%) during a repeat
cycle of one week, the influence on the accuracy of the Mie winds acquired over the entire globe is well below 0.1 m·s$^{-1}$. However, a more noticeable contribution of the high frequency noise to the Mie wind bias is expected in regions where the reaction wheels are run at the critical speeds, although further studies are necessary for verification. The Rayleigh wind results are hardly affected, most likely because other noise contributors dominate, such as photon shot noise and fluctuations of the incidence angle on the spectrometers. Concerning the ground velocities which are only collected in areas with high surface
albedo, a similarly small impact of the laser frequency stability was determined. The mean ground-return velocity derived from the Mie spectrometer changes linearly with the frequency stability on observation level by about 0.08 m·s$^{-1}$·MHz$^{-1}$, whereas the Rayleigh ground velocities are nearly independent of the frequency noise over the investigated range.

Despite the small effect on the wind data, application of a QC in the Aeolus processor could be foreseen for the Mie winds to filter out measurements during which the frequency stability is worse than 20 MHz. The same threshold is suggested for the
processing of the Mie and Rayleigh ground velocities, as this will slightly improve the accuracy of the ground velocity values while their number is not considerably reduced. These QC approaches become more important in case the atmospheric return SNR is further decreasing within the mission lifetime, as this will increase the share of Mie wind results among the overall Aeolus wind observations. For the Rayleigh winds, a QC based on the frequency stability is not considered useful, since the error is largely dominated by shot noise. Hence, the discarding of any measurements, even with poor frequency stability, will
rather diminish the Rayleigh wind precision given that it is primarily limited by shot noise.



Concerning the use of Aeolus data on smaller geographical scales, e.g. for its validation by on-ground or airborne instruments, the geolocational dependence of the laser frequency stability should be considered. In regions where the reactions wheels repeatedly rotate at their critical frequencies suffer from enhanced frequency noise during every overpass along Aeolus' ascending or descending orbit. Consequently, the percentage of wind observations with degraded stability is considerably

higher than on a global scale and is expected to have a stronger impact on the wind accuracy, especially for the Mie channel. With a view to future space lidar missions, particularly those which have strict requirements in terms of the laser frequency stability like EarthCARE, ACDL, MERLIN or Aeolus follow-on missions, several lessons learnt are derived from the current study: First, the cavity control scheme of the laser transmitter plays an important role to ensure a sufficiently short dead time of the feedback loop, and in turn, to filter out vibration frequencies of several hundreds of Hz which potentially deteriorate the

laser stability. A modified ramp-delay-fire technique that was developed for the A2D (Lemmerz et al., 2017) allows for fast responses in the microsecond-regime, thus providing high frequency stability (<4 MHz RMS) even in harsh vibration environments. Second, although the on-ground micro-vibration tests led to the important implementation of isolation suspensions which effectively attenuated the micro-vibration levels by about one order of magnitude, the tests did not clearly reveal the existence of the critical reaction wheel speeds which would affect performance in a measurable way. This is mainly

due to the too short operation of the wheels at their critical speeds which did not allow for sufficient statistics of the acquired laser data depending on wheel speed. Therefore, extended tests with the spacecraft mounted in an isolated configuration and with flight-representative time-varying reaction wheel speeds whilst operating the laser are recommended. Moreover, it is proposed to use an appropriate number of accelerometers at different locations of the laser bench during such tests to properly characterize the impact of micro-vibrations on the laser behavior, particularly the frequency stability. Finally, a higher temporal

resolution of the laser housekeeping telemetry data than currently available for Aeolus (0.25 Hz) with a focus on cavity control parameters is advisable to better evaluate the laser performance both on-ground and in space.

Furthermore, it is considered that the installation of accelerometers in the instrument flight model and the acquisition of micro-vibration measurements in orbit, even resulting in modest amounts of data, is extremely beneficial. These not only will allow a more definitive identification of the spacecraft elements responsible for deviations scientific measurements, but will also

provide a reference for correlation of mechanical models representing orbital conditions.

Passive suspensions provide a well-proven solution for the mitigation of micro-vibrations generated by reaction wheels in orbit. With their simple configuration, they have low mass, consume no electrical power and require no computational resources. However, inherently to their mechanical architecture, they produce a non-negligible amplification of the disturbances at the frequencies of the suspension modes. Also, extreme isolation at high-frequencies, which can be required

by some applications, is often compromised by the presence of spurious mechanical modes whose effects in orbit are difficult to predict. As an alternative, active closed-loop vibration isolation systems based on sensing and counteracting the mechanical disturbances arising in the spacecraft show a great promise in overcoming the limitations of passive systems. These can be implemented at the level of both the source (Preda et al., 2018) and/or the payload (Sanfedino et al., 2020) and can significantly mitigate the micro-vibration environment in Aeolus follow-on and other future space lidar missions.



*Data availability.* The presented work includes preliminary data (not fully calibrated/validated and not yet publicly released) of the Aeolus mission that is part of the European Space Agency (ESA) Earth Explorer Programme. This includes wind products from before the public data release in May 2020 and/or aerosol and cloud products, which have not yet been publicly released. The preliminary Aeolus wind products will be reprocessed during 2020 and 2021, which will include in particular a significant L2B product wind bias reduction and improved L2A radiometric calibration. Aerosol and cloud products will become publicly available by spring 2021. The processor development, improvement and product reprocessing preparation are performed by the Aeolus DISC (Data, Innovation and Science Cluster), which involves DLR, DoRIT, ECMWF, KNMI, CNRS, S&T, ABB and Serco, in close cooperation with the Aeolus PDGS (Payload Data Ground Segment). The analysis has been performed in the frame of the Aeolus Data Innovation and Science Cluster (Aeolus DISC). The raw Mie channel data used for the assessment of the laser frequency stability is included in the Aeolus L1A product which was provided from the Aeolus Online Dissemination Service ADDF. The housekeeping telemetry data were obtained from the Aeolus MUST tool which is accessible for members of the Aeolus Data Science and Innovation Cluster (DISC). Aeolus L2B and ECMWF model winds were extracted from the L2C product which is available from the tool *VirES for Aeolus* (https://aeolus.services/). The ground velocity data is part of the reprocessed Aeolus AUX_ZWC product and will become publicly available in 2021.

*Competing interests.* The authors declare that they have no conflict of interest.

*Author contribution.* OL analyzed the ALADIN laser frequency stability, its correlation with the reaction wheel speeds and its impact on the Aeolus wind data quality. CL, FW, TK, DW and OR supported the data analysis and interpretation of the results. GR and AH studied the platform parameters related to the Aeolus attitude and orbit control system. GR investigated the results from the on-ground micro-vibrations tests with regards the in-orbit instrument performance. OLe, PM and FF were involved in the on-ground micro-vibration tests of the ALADIN instrument and provided insight to the test procedure and findings at that time. PB provided information of the ALADIN MO cavity control scheme. TP is the Aeolus Mission Manager. OR is scientific coordinator of the Aeolus DISC. The paper was written by OL with contributions from all co-authors.

*Acknowledgements.* The Doppler wind lidar ALADIN was built by Airbus SAS in Toulouse, France, the satellite by Airbus Ltd in Stevenage, UK, and the laser transmitters by Leonardo S.p.a. in Florence and Pomezia, Italy. The authors acknowledge Anne Grete Straume (Aeolus Mission Scientist) and Jonas von Bismarck (Aeolus Data Quality Manager) as well as the Aeolus Mission Advisory Group, the Aeolus Space and Ground Segment Operations teams and the Aeolus DISC for their invaluable contributions. The authors are also grateful for the insightful discussions with colleagues from ESA's SWARM mission, particularly Nils Olsen, Anja Strømme, Rune Floberghagen, Laurent Maleville and Jerome Bouffard, regarding the interaction between the reaction wheels, magnetorquers and magnetometers on-board Aeolus which helped to rule out a direct influence of the Earth magnetic field on the laser frequency stability. The first author was partly funded by a young scientist grant by ESA within the DRAGON 4 program (contract no. 4000121191/17/I-NB).



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
