# Peer review of "ALADIN laser frequency stability and its impact on the Aeolus wind error"

_Atmospheric Measurement Techniques, 2021_

## Author Comment (AC1)

***Response to Referee Comment (RC1) on***

*ALADIN laser frequency stability and its impact on the Aeolus wind error*

*(https://doi.org/10.5194/amt-2021-74)*

We thank the reviewer for reading our manuscript and for his positive feedback. Our response to his comment is provided below.

General comment:

*The long-term laser frequency stability is first reported for spaceborne high energy solid-state laser. The difference performance between under-ground and in- orbit is implemented. The enhanced frequency noise due to the satellite's reaction wheels is discovered. Two year's global frequency stability of laser is present. It is very significant for future frequency-stability spaceborne laser development. Aeolus wind error in both Mie and Rayleigh Channel due to the enhanced frequency noise is analysed. The wind error can be accepted for the ECMWF mode.*

Comment #1:

*It is better that the mechanics of the frequency noise enhancement in the master oscillator due to micro-vibration is given.*

Response to Comment #1:

We are not sure whether we understand the reviewer's comment regarding the mechanics of the frequency noise enhancement correctly. The underlying mechanical process that causes the enhanced laser frequency noise is explained in line 75ff.:

> "With regard to Aeolus the main susceptibility to micro-vibrations is related to the alteration of the laser cavity length which leads to frequency fluctuations of the emitted light."

Variations in the MO cavity length as being the root cause for frequency fluctuations are also discussed in the description of the ALADIN laser frequency stability in section 3.1, e.g, in lines 280ff:

> "It should be pointed out that, apart from laser frequency variations caused by cavity length changes, the measured Mie response […]".

Additional information on the mechanical and optical layout of the ALADIN laser transmitter, particularly the design of the folded master oscillator, is provided in the ESA Science Report to the Aeolus mission (ESA, 2008), p. 60:

> "The different stages shown in the PLH architecture are divided into two optical benches in the actual laser head: the Upper Optical Bench, and the Lower Optical Bench inside the laser housing. […] The UOB carries the cold plate, which allows cooling of the active components (Master Oscillator and the Pre- and Power Amplifiers). It also carries the isostatic mounts, which fix the PLH onto the ALADIN structure. These isostatic mounts have to maintain alignment of the output beam with respect to the ALADIN optics under the varying forces acting on the cold plate. […] The folding mirrors of the Master Oscillator are mounted on an Invar substructure for additional stability."

For the sake of conciseness, we did not mention these details in the text and referenced the available literature at the beginning of section 2.1.

---

## Author Comment (AC2)

***Response to Referee Comment (RC2) on***

*ALADIN laser frequency stability and its impact on the Aeolus wind error*

*(https://doi.org/10.5194/amt-2021-74)*

We appreciate the referee's very insightful and helpful remarks on our manuscript. The responses to the individual comments and the corresponding changes that will be made to the manuscript are presented in the following.

General Comment:

*This paper provides a history and analysis of the on-orbit Aeolus frequency stability, its relationship to the satellite reaction wheel rotation velocities, and the impact on the wind measurements. It must have been very interesting to discover the relationship and to be able to clearly characterize it. The paper describes the relationship between specific reaction wheel speeds and laser frequency stability and the follows the issue all the way through to the impact different Mie and Rayleigh channel performance (accuracy and random error) for both atmospheric and ground returns. The paper conclusion section includes a summary of lessons learned and several important suggestions for mitigating this issue on future missions that require frequency stabilized lasers.*

Response to General Comment:

We are very grateful for the referee's thorough reading of our paper, and the valuable and positive comments. It was indeed intriguing to discover the impact of the reaction wheels on the laser frequency stability, especially as we initially followed a "red herring" by investigating the correlation between the frequency fluctuations and the magnetometer data (see Comment #22 below).

Comment #1:

*The paper is somewhat unusual for a scientific paper as there are few equations (143) and few variables, however the authors do a good job of explaining, with excellent graphics, what may be considered a complex issue for some readers. The paper includes a detailed description of the Aeolus instrument as well as information on the Mie retrieval, and Instrument Response Calibration step. One wonders whether this information is well covered in another paper in this special issue of AMT that could be referred here. If not, this paper will serve as a great reference for that material.*

Response to Comment #1:

We agree with the reviewer that the paper is quite comprehensive and contains detailed information about the ALADIN instrument and its measurement modes. Although the information is partly covered in other articles of the AMT special issue, we think that the degree of detail is appropriate to ensure understandability of the complex relationships presented in the manuscript. Also, we hope that the paper will reach a broad readership beyond the Aeolus community which may not be familiar with the specific features of ALADIN. In this context it should be noted that an overview paper on ALADIN and its performance is still outstanding, but the instrument and its modes are well described in the L1B Algorithm Theoretical Basis Document (Reitebuch et al., 2018).

Comment #2:

*The paper is well organized, thorough, and provides an important contribution to the field. Thus, it is difficult to come up with any major issues or recommendations for improving the paper other than perhaps shortening it a bit. Some messages are relayed in multiple ways, likely in an effort to educate and convince the reader, Perhaps, some of the graphics and corresponding explanation of the relationship between the reaction wheel speeds and laser frequency stability could be put into an appendix, but this is only necessary if the journal imposes page limitations.*

Response to Comment #2:

Thanks a lot for the positive appraisal of our paper. Despite the occasional repetition of some of the findings in the text, we think that shortening the manuscript bears the risk of impairing the comprehensibility of the manuscript. Since the latter covers both technical (laser design, reaction wheels, micro-vibrations) and scientific aspects (frequency stability, Aeolus wind error), different groups of readers with diverse prior knowledge of discussed issues are addressed. For this reason and given the fact that there are no page limitations imposed by the journal, we prefer to keep the structure of the manuscript in the current form.

Comment #3:

*While not necessary (and likely not appropriate) for this paper, it would be nice to know if a structural/modal analysis of the laser bench has been provided by the laser vendor that demonstrates the expected laser frequency stability levels at the reaction wheel speeds/frequencies where sensitivity was observed on orbit. Likewise, it would be interesting to know if there was any relationship between the reaction wheels and the performance of the interferometer spectrometers (or alignment between them), but again, this is not needed for this paper. Overall, the paper is excellent and highly recommended for publication.*

Response to Comment #3:

Extensive micro-vibration verification activities were carried out prior to the launch of the satellite, as mentioned in section 3.4 of the manuscript. This included both micro-vibration tests at laser level with representative mechanical excitation spectra as well as micro-vibration tests at satellite-level with the reaction wheels being operated over their entire operational speed ranges. These tests demonstrated that, despite the fact that the peak of disturbances from the reaction wheels coincided with the most susceptible frequencies of the lasers (mainly in the 400 to 600 Hz frequency band), the vibration levels were lower than the danger-levels previously established to them.

The micro-vibration environment on the instrument induced by the reaction wheels results from the imperfections on the cage and bearings, the resonance frequencies of the rotor and cage, their coupling to the stiffness of the wheels mounting brackets, and the resonant frequencies of the platform as well as those of the instrument. Therefore, significant discrepancies can be expected between the spectra of disturbance characterized on the wheels stand-alone, and the spectra of the micro-vibration environment measured on the instrument.

A future path of investigation could pass by identifying the main contributors to the degradation of performance of the instrument, whether normal modes of the instruments which result in misalignment between optical elements or the excitation of critical frequencies of some units, such has the internal cavities of the laser heads. Correct identification of the overall satellite coupled behaviour and micro-vibration levels on the instrument can only be performed by dedicated tests, since the accuracy of finite elements is gradually lost for increasing frequencies. Even testing a complete satellite on ground for micro-vibrations is extremely challenging because structural damping and joint-dependant stiffness and natural frequencies can be heavily affected by gravity and support boundary conditions.

Comment #4:

*Line 28-30: This sentence is confusing: "Hence, although the Mie wind bias is increased by 0.3 $m \cdot s^{-1}$ at times when the frequency stability is worse than 20 MHz, the small contribution of 4% from all wind results renders this effect insignificant ($<0.1 m \cdot s^{-1}$) when all winds are considered. - What is the source of the 4%? "from all wind results"? All Mie winds or Mie and Rayleigh winds? Do the authors mean that the 0.3 m/s bias during frequency instability periods is dwarfed by other sources of random and accuracy errors?*

Response to Comment #4:

The percentage refers to the total amount of valid Mie winds that were measured over the investigated periods of one week. As pointed out in section 4.1, the distributions of the frequency stability for the two data subsets (only measurements containing valid Mie or valid Rayleigh winds) are almost identical although the number of valid Mie winds is usually about three times lower than the number of valid Rayleigh wind on a global scale. Hence, the contribution of measurements with enhanced noise (>20 MHz) is the same, regardless whether all wind results (Mie and Rayleigh), or only Rayleigh winds are considered. Nevertheless, in order to avoid confusion, we will change the sentence in the abstract as follows:

> "Hence, although the Mie wind bias is increased by 0.3 m·s$^{-1}$ at times when the frequency stability is worse than 20 MHz, the small contribution of 4% from all Mie wind results renders this effect insignificant (<0.1 m·s$^{-1}$) when all winds are considered."

In this sense, it is true that the bias change introduced by the frequency instability is hidden by larger error sources and becomes only visible by filtering for measurements with considerably enhanced frequency noise.

Comment #5:

*Line 33-34: I found the meaning of this sentence difficult to pull out until reading the corresponding section, perhaps because of the term "sorts out"? Are the authors implying that, "Even if one considers only time periods of data with >20 MHz frequency stability, the impact on accuracy of the Mie and Rayleigh ground velocities is still less than 0.15 m·s$^{-1}$." (?)*

Response to Comment #5:

This sentence is indeed misleading since the results in section 4.2 (Fig. 15) show that discarding those ground velocities for which the frequency stability is worse than 20 MHz changes the mean ground velocity by only 0.05 m·s$^{-1}$, namely from about 0.15 m·s$^{-1}$ (right blue data point in Fig. 15 (a), no observations filtered out) to about 0.10 m·s$^{-1}$ (at a threshold of 20 MHz). When a stricter threshold of 10 MHz is applied, the accuracy is only improved by a bit more than 0.10 m·s$^{-1}$. Thus, the sentence in the abstract will be changed to

> "Here, the application of a frequency stability threshold that filters out wind observations with variations larger than 20 or 10 MHz improves the accuracy of the Mie and Rayleigh ground velocities by only 0.05 m·s$^{-1}$ and 0.10 m·s$^{-1}$, respectively, however at the expense of useful ground data."

Comment #6:

*Paragraph starting around line 60: Not all Doppler lidar and HSRL techniques require frequency stability. See Bruneau and Pelon 2021 (https://amt.copernicus.org/articles/14/4375/2021/) and Bruneau et al. 2013, https://www.osapublishing.org/ao/abstract.cfm?uri=ao-52-20-4941.*

Response to Comment #6:

Agreed. The paragraph will be revised as follows:

> "The laser frequency stability is a crucial parameter for the Aeolus mission and many Doppler wind lidar instruments in general […]. However, it should be mentioned that not all Doppler lidar and HSRL techniques require high frequency stability, especially if the referencing to the outgoing signal is performed on a pulse-to-pulse basis (Baidar et al., 2018; Tucker et al., 2018; Bruneau and Pelon, 2021)."

Comment #7:

*Line 68: Perhaps clarify "beyond" 1 Hz" as frequencies "greater than 1 Hz?*

Response to Comment #7:

The sentence will be changed accordingly.

Comment #8:

*Line 74-80: The authors might also consider including the work done for the GRACE-follow-on mission, https://www.repo.uni-hannover.de/bitstream/handle/123456789/10524/PhysRevLett.123. 031101.pdf?sequence=1*

Response to Comment #8:

Thanks for pointing us to this work which will be referenced in the following sentence in the introduction of the revised manuscript:

> "Regarding the Gravity Recovery and Climate Experiment (GRACE) Follow-On mission, the sensitivity of the Laser Ranging Interferometer instrument at frequencies greater than 0.2 Hz is limited by the frequency stability of the laser which was assessed by ground tests prior to the launch in 2018 (Abich et al., 2019)."

Comment #9:

*Figure 1 – this is one of the better block diagrams/optical schematic that I've seen for lidar systems. It is clear and helps the reader understand the optical path. It does seem familiar, however. If it has been used in other papers by this group, it may be good to reference the previous document in the caption as, "Figure after XXX et. al., 2019" (or whatever is appropriate).*

Response to Comment #9:

The figure was created by combining the information from several sources that describe the experimental setup of the ALADIN instrument and its airborne prototype, the ALADIN Airborne Demonstrator. We are not aware that this or a similar schematic was already used in other publications. Therefore, it is not necessary to add a reference to the caption.

Comment #10:

*Line 134-139: Suggest replacing "UV emit beam" with "transmitted UV beam" (keeping with "laser transmitter" on line 105-106). Likewise clarify laser radiation (vs. radiation) or use just one term (transmit beam).*

Response to Comment #10:

We decided to simply write "UV beam" to avoid confusion. The term emit beam was initially used to be consistent with the so-called "emit path" which describes the section of the instrument between the laser and the telescope. In addition, we will replace "(laser) radiation" with "beam".

Comment #11:

*Also – why 0.5%? Why not a smaller amount since it's being attenuated anyway? Typically, T0 reference beams are "leaked" through a 99.9% reflectivity mirror, and still require attenuation.*

Response to Comment #11:

The beam splitter is indeed realized by a highly-reflective mirror whose reflectance does however not exceed 99.5%. The number of dielectric layers was chosen as a compromise of high reflectance and high damage threshold in the UV spectral region (given that higher reflectance requires a larger number of dielectric layers).

Comment #12:

*Line 182 – "on-ground" or "pre-flight" (On-ground could imply there's a ground-version of the laser being used for test).*

Response to Comment #12:

This will be changed accordingly.

Comment #13:

*Line 192-194 – What was the amplitude of the vibrations? (order of magnitude)*

Response to Comment #13:

Unfortunately, we do not have this information. As reported in the paper by Mondin and Bravetti (2017), "since it was not possible to interrupt the vacuum during the on-going test measurements, the accelerometer was positioned for a purely qualitative measurement on the door of the TVC" and the acquired data were never calibrated.

Comment #14:

*Line 230 – What is the "PD"? (is it the MO photodiode?)*

Response to Comment #14:

Yes, this will be clarified in the text.

Comment #15:

*Line 237 – May wish to say, "As of the writing of this paper, the energy has remained above 60 mJ."*

Response to Comment #15:

We will change the text accordingly.

Comment #16:

*Line 271-272 – Just curious, why wasn't the more "agile" ramp-fire technique used on A2D also used on Aeolus?*

Response to Comment #16:

The refined ramp-fire technique used on the A2D was developed around 2017 when the design Aeolus laser and particularly the cavity control scheme of the MO was already finalized.

Comment #17:

*Figure 4 – the colored dashed lines are a little difficult to see, especially the green one (I had to zoom in to see it after reading about it in the text). Can you make these slightly bolder?*

Response to Comment #17:

The coloured dashed lines will be made thicker and the green line will be made brighter.

Comment #18:

*Line 368 – how does "opacity" come into the display? Aren't all the dots opaque, with varying color from white to dark red?*

Response to Comment #18:

Although it might be hard to recognize in Fig. 6, the dots show indeed different opacity which scales with the displayed frequency stability.

Comment #19:

*Figure 5 and corresponding text –*

- *This is a good figure, demonstrating the information clearly in two different ways*
- *What is a "frequency step"? The term is only used in this figure caption, but not defined.*
- *Previous discussion used as 12 s analysis, what was the 24s?*
- *Here, and previously, the authors use phrases such as "standard deviation of the relative frequency on pulse-to-pulse level within this period" or "frequency stability in terms of the standard deviation of the relative frequency over the 540 pulses within that observation." Perhaps back on line 268 the authors could introduce a variable name such as, ".the standard deviation of the relative frequency over one observation (540 pulses, 12 s), which we'll refer to hereforth as sf(N=540)." Then they authors can use this term instead of the lengthy phrases.*

Response to Comment #19:

The temporal scheme of the IRCs will be clarified as follows:

> "The procedure involves a frequency scan over 1 GHz in steps of 25 MHz to simulate well-defined Doppler shifts of the atmospheric backscatter signal within the limits of the laser frequency stability. During the IRC which takes about 16 minutes, two observations (each 12 s) for each of the 40 frequency steps, the contribution of […]".

Moreover, we will use the suggested variable to avoid the repetition of lengthy descriptions.

Comment #20:

*Figure 8 – This is a very interesting figure. It must have been an exciting discovery to see these results for the first time.*

Response to Comment #20:

It absolutely was!

Comment #21:

*Line 452-453 (plus Tables 2,3 and surrounding text) – The authors state, "The variability in the center frequency of the common critical wheel speeds is on the order of 0.1 RPS which is comparable to the average width of the fitted peaks." Is this actual variability in the reaction wheel speed, or is it uncertainty in the knowledge of the speed? Do the vendors provide an uncertainty on the knowledge of the reaction wheel speeds? This isn't a critical point to be addressed, just a matter of curiosity.*

Response to Comment #21:

The reaction wheel speed is known with an accuracy of about 0.5 RPM or 0.0083 RPS, respectively. Thus, the determined variability in the centre frequency of the common critical wheel speeds is not limited by the uncertainty in the knowledge of the speed. The following sentence will be added for clarification:

"Note that the individual wheel speeds are known with an accuracy of about 0.01 RPS."

Comment #22:

*Line 537-543 - this paragraph hints that there may be more to the magnetometer-laser frequency instability relationship than just an indirect relationship through the reaction wheel speeds, but the issue isn't explored further. It comes up again in the conclusions, leaving the reader wondering if there might more to the story. Perhaps here the authors could clarify either that they have established there is not an expected impact of the magnetic fields on the laser frequency, or that this is an area open to further study.*

Response to Comment #22:

As a matter of fact, the investigation of the geolocational patterns shown in Fig. 5 initially focussed on magnetometer data, yielding a decent correlation between the laser frequency stability and the on-board measured magnetic field intensity, especially for descending orbits. This led to the assumption that the geomagnetic field might impact the laser frequency stability, e.g., by introducing changes in the length of the laser bench via magnetostriction.

Further analysis of the magnetometer data, however, revealed a strong influence from the magnetic fields generated by the magnetorquers on-board Aeolus which serve the regulation of the reaction

wheel speeds. The strength of the magnetic field produced by the three magnetorquers at the location of the magnetometer was estimated to be about as high as that of the geomagnetic field intensities (tens of μT). The magnetic field measured by the magnetometers hence represents a superposition of geomagnetic and magnetorquer fields. The reaction wheel speeds follow a roughly periodic profile over the orbit due to a consistent periodic environmental disturbance torque profile, dominated by aerodynamic drag, and the periodic effectiveness of the magnetorquers. Since the latter is a function of the magnetic field direction and strength, there is an indirect and complex relationship between the critical wheel speeds, and hence the frequency stability, and the magnetometer data via the superimposed magnetic fields of the magnetorquers and the Earth.

In view of the already long manuscript, we decided to omit these explanations, even though they were important findings that led us on the right track, namely the investigation of reaction wheel speeds over periods of one week.

Comment #23:

*Line 678 – Perhaps the authors could explain here why the Rayleigh wind accuracy is less impacted by the frequency stability? The question is briefly addressed in the conclusion (lines 825-830). Could the difference also be attributed to the number of Rayleigh (vs. Mie) observations that may average out biases?*

Response to Comment #23:

As mentioned in the manuscript, the main reason why the Rayleigh channel is less sensitive to frequency fluctuations is the fact that the Rayleigh wind error is much more dominated by shot noise than the Mie wind error. The number of observations does not play a role since the percentage of wind results for which the frequency noise is enhanced is almost identical for both channels (see also Response to Comment #4).

Comment #24:

*Line 727-729: The authors may wish to explain briefly what is meant by "atmospheric contamination." (Such a term seems ironic, given Aeolus' mission to measure the speed of the atmosphere, but I digress). Presumably, it means that within a finite sized range bin they want the signal to be dominated by ground return, and so high albedo surface observations are chosen. That said, is there a concern about the impact of blowing snow over the arctic/Antarctic surfaces on the retrieval?*

Response to Comment #24:

That's exactly right. The term atmospheric contamination describes the detrimental impact of the broadband atmospheric backscatter return signal on the narrowband ground return signal which can alter the retrieved ground velocity as a systematic error. This effect is especially strong when the ground signal is low, i.e., over low-albedo surfaces, and when the vertical extent of the ground is small compared to the bin thickness. In addition, blowing snow over Arctic/Antarctic surfaces leads to a narrowband Mie return with a potentially non-vanishing Doppler shift, thus acting as another error source on the ground returns. Since the LOS pointing and blowing snow direction could randomly vary over a larger dataset, the latter effect is rather considered as a source of random errors, but more studies are needed on this topic. Both effects that are differently strong over different geolocations are indeed impairing the accuracy and precision of the ground velocity and represent bigger issues than the frequency stability.

The text will be revised as follows:

"Given the susceptibility to atmospheric contamination, i.e., the detrimental impact of the broadband atmospheric backscatter return signal from Rayleigh scattering on the narrowband ground return signal, only ground velocities that were measured from surfaces with high albedo in the UV spectral range are considered valid, which drastically limits the number of available ZWC values. […] Here, the precision of the ground returns is impaired by blowing snow that leads to a narrowband Mie return with a potentially non-vanishing Doppler shift, thus acting as another source of random errors on the ground returns."

One of the conclusions in section 4.2 will be extended as well:

"Although the impact is slightly larger, it is still hardly noticeable in the statistics derived from the complete ZWC dataset and certainly represents a smaller issue than e.g., blowing snow affecting the Mie and Rayleigh responses."

Comment #25:

*Lines 808-810 – see previous comment about the magnetometer discussion.*

Response to Comment #25:

See Response to Comment #22.

Comment #26:

*Lines 840-841 – Why was this technique implemented on Aeolus? If it was based on technology readiness for space, has that changed since launch?*

Response to Comment #26:

The choice of the locking technique was made in the early project development phase and was based on the heritage of the development team. At that time, the suitability of the choice was demonstrated and the electronics was qualified for space. In the meanwhile, other methods have reached a much higher TRL and could be considered for replacement in future missions.

Comment #27:

*Page 34 – When listing recommendations for mitigating the issue on future missions, do any future mission concepts include the ability to reference (adjust) the measurement on a pulse by pulse basis prior to pulse accumulation? I understand this is not feasible for the ACCD detection approach, but what about other future ESA-funded lidars?*

Response to Comment #27:

Referencing on a pulse-by-pulse basis, as performed in other systems basis (Baidar et al., 2018; Tucker et al., 2018; Bruneau and Pelon, 2021), would indeed relax the requirements in terms of frequency stability. The new generation of Aeolus and also ATLID onboard EarthCARE will however also include accumulation CCDs. The acquisition on a pulse-by-pulse basis could be possible with a detector working in real counting mode. There are studies ongoing with avalanche photodiodes in analogue mode that offer very low dark noise. However, in the current state of development, the SNR obtained with these APDs is still be too poor to determine the frequency from the atmospheric backscatter signals of single pulses, except for the ground return perhaps. The following paragraph will be added to the conclusions section of the manuscript:

> "Referencing on a pulse-by-pulse basis, as performed in other wind lidar instruments (Baidar et al., 2018; Tucker et al., 2018; Bruneau and Pelon, 2021), would relax the requirements in terms of frequency stability. This could, for instance, be realized with avalanche photodiodes (APDs) working in real counting mode. There are studies ongoing with APDs in analogue mode offering very low dark noise. However, in the current state of development, the SNR obtained with these APDs is still too poor to determine the frequency from atmospheric backscatter of single pulses, so that signal accumulation is expected to be necessary for most future space (wind) lidar missions."

Comment #28 (General editing suggestions):

*While the paper is quite well written, the following is a list of suggestions that could be used to improve the grammar. They should not be considered necessary for paper publication.*

- *A comma belongs after use of "i.e." or "e.g.", ("e.g., as shown in this example")*
- *Line 82: replace "...spectrometers which allow to assess this…" with "...spectrometers that enable assessment of this…"*
- *Line 100: replace "This chapter will provide a brief description of the ALADIN instrument and its operating principle." With, "This section provides a brief description of the ALADIN instrument and its operating principle." The rest of the paragraph is in present tense, and the section (not a book chapter) is already providing the information.*
- *Line 109: It might not be clear to all readers what "switchable" means so I suggest replacing "The two fully redundant laser transmitters, referred to as flight models A and B (FM-A, FM-B), are switchable by a flip-flop mechanism (FFM)." with, "A flip-flop mechanism provides the ability to switch between the two fully redundant laser transmitters, referred to as flight models A and B (FM-A, FM-B)."*
- *Line 173-174: suggest replacing "This is especially true, as the atmospheric backscatter signals from multiple outgoing laser pulses are accumulated to measurements before data down-link." with "This is especially true, as atmospheric backscattered signals from multiple outgoing laser pulses are accumulated on the CCD prior to digitization and data down-link.*
- *Line 303: Suggest replacing, "Nevertheless, there is also a considerable amount of observations (19%) for which the frequency stability better than 5 MHz, i.e. comparable to the A2D laser performance" with "However, there are also a considerable number of observation periods (19%) for which the frequency stability is better than 5 MHz, i.e., comparable to the A2D laser performance"*
- *Line 425: ", which can be attributed to being located further away in the instrument" - do you mean RWA3 is located further away from the instrument or laser FMA is further away in the instrument?*
- *Line 431: suggest "...greater disturbing effect on one laser or another due to being located closer-by."*
- *Line 535-536: suggest "As a result, the Allan deviation on the observation level is around $(0.7 \pm 0.1)$ MHz almost independent of enhanced noise periods."*

- *Line 559-561: suggest "These included disturbing the laser transmitter with representative mechanical excitation spectra, thus identifying susceptibility in the 400 Hz to 600 Hz frequency band, as well as around 250 Hz."*

- *Figure (12) – the term "frequency stability on measurement level" is used a few times in this figure (and supporting paragraph) –*

- *Line 739 AND 768: suggest replacing "In analogy to…" with "Analogous to…" or "By analogy with…"*

- *Line 832 – remove "In" from "In regions…" and start the sentence with "Regions…"*

Response to Comment #28:

We thank the reviewer for his meticulous reading and comments which helped to improve the grammar of the manuscript. All suggested revisions will be made to the revised manuscript.